# CMPhysBench: A Benchmark for Evaluating Large Language Models in Condensed Matter Physics

**Weida Wang[1,4*] Dongchen Huang[2,3*] Jiatong Li[6*] Tengchao Yang[1,5*] Ziyang Zheng[2,3*],**
**Chuyi Peng[2,3], Di Zhang[4], Dong Han[1], Benteng Chen[1], Binzhao Luo[2], Zhiyu Liu[2,3], Kunling Liu[2,3],**
**Zhiyuan Gao[2,3], Shiqi Geng[1], Wei Ma[5], Jiaming Su[5], Xin Li[5], Shuchen Pu[1], Yuhan Shui[1],**
**Qianjia Cheng[1], Zhihao Dou[1], Dongfei Cui[1], Changyong He[5], Jin Zeng[5],**
**Zeke Xie[8], Mao Su[1], Dongzhan Zhou[1], Yuqiang Li[1], Wanli Ouyang[1], Yunqi Cai[2,3†],**
**Xi Dai[7†], Shufei Zhang[1†], Lei Bai[1†], Jinguang Cheng[2†], Zhong Fang[2†], Hongming Weng[2,3†]**

[1]Shanghai Artificial Intelligence Laboratory
[2]Beijing National Laboratory for Condensed Matter Physics and Institute of Physics, Chinese Academy of Sciences
[3]Condensed Matter Physics Data Center, Chinese Academy of Sciences
[4]Fudan University  [5]Tongji University  [6]Hong Kong Polytechnic University
[7]Hong Kong University of Science and Technology
[8]Hong Kong University of Science and Technology (Guangzhou)
`{wangweida,zhangshufei,bailei}@pjlab.org.cn,`
`{huangdongchen,caiyq,jgcheng,zfang,hmweng}@iphy.ac.cn, daix@ust.hk`

## Abstract

We introduce CMPhysBench, designed to assess the proficiency of Large Language Models (LLMs) in **C**ondensed **M**atter **Phy**sics, as a novel **Bench**mark. CMPhys-Bench is composed of more than 520 graduate-level meticulously curated questions covering both representative subfields and foundational theoretical frameworks of condensed matter physics, such as magnetism, superconductivity, strongly correlated systems, etc. To ensure a deep understanding of the problem-solving process, we focus exclusively on calculation problems, requiring LLMs to independently generate comprehensive solutions. Meanwhile, leveraging tree-based representations of expressions, we introduce the Scalable Expression Edit Distance (SEED) score, which provides fine-grained (non-binary) partial credit and yields a more accurate assessment of similarity between prediction and ground-truth. Our results show that even the best models, Grok-4, reach only 36 average SEED score and 29% accuracy on CMPhysBench, underscoring a significant capability gap, especially for this practical and frontier domain relative to traditional physics. The code and dataset are publicly available at https://github.com/CMPhysBench/CMPhysBench.

## 1 Introduction

Recent advances in large language models (LLMs) have revolutionized natural language processing, demonstrating exceptional capabilities in understanding and generation tasks (Brown et al., 2020; Devlin et al., 2019), particularly in commonsense and mathematical reasoning, often enhanced by reinforcement learning techniques (Guo et al., 2025; Kojima et al., 2022). Leveraging these strengths, LLMs have achieved impressive results in Olympiad-level mathematics (Zhang et al., 2025a), complex programming (El-Kishky et al., 2025), and even scientific discovery (Bai et al., 2025; Yang et al., 2023; Wang et al., 2025), fueling expectations for their applicability in physics. As a field grounded in uncovering the fundamental laws of nature, physics imposes uniquely rigorous demands on LLMs, requiring not only advanced reasoning and mathematical precision but also a deep conceptual understanding of physical principles, concepts and approximations making it an ideal testbed for evaluating whether LLMs truly comprehend the structure of the real world.

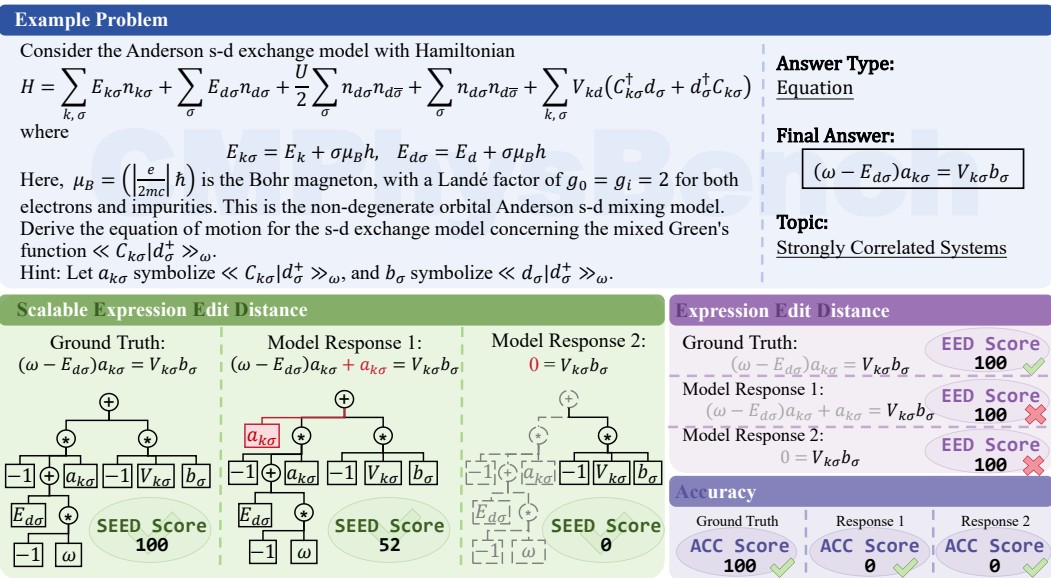

Figure 1: Example problem from CMPhysBench comparing three metrics for model performance evaluation: Expression Edit Distance (EED) (Qiu et al., 2025b), Accuracy (Acc) (He et al., 2024) and the proposed Scalable Expression Edit Distance (SEED). Scores for three different responses to the same problem are shown, where SEED excels at both accuracy and fine-grained evaluation. The detailed interpretation of symbols is shown in Appendix H.

Previous benchmark efforts, such as SciQ (Welbl et al., 2017) and ScienceQA (Saikh et al., 2022), have played an important role in facilitating the evaluation of LLMs on physics-related questions. However, these benchmarks primarily focus on high school-level content, which may not adequately test the complexity of reasoning or the degree of mathematical rigor required for evaluating advanced understanding in physics. More recent benchmarks, including PHYBench (Qiu et al., 2025b) and UGPhysics (Xu et al., 2025a), have made meaningful progress by incorporating undergraduate-level problems. Nonetheless, these benchmarks remain limited in depth, as they often underrepresent the most critical and frontier areas of contemporary physics research. Considering the inherent conceptual and mathematical complexity of physics, *broader and more rigorous benchmarks are essential for assessing whether LLMs can support real-world scientific tasks and facilitate cross-disciplinary integration.*

In this work, we focus on Condensed Matter Physics (CMP), which becomes the mainstream of current physical research and investigates the physical properties and microscopic structures of condensed phases of matter, namely solids and liquids (Marder, 2010). As a central area of modern physics, condensed matter has become a driving force behind many recent theoretical and experimental advances, contributing to our understanding of phenomena such as superconductivity, topological states, and quantum phase transitions. This field integrates concepts from quantum mechanics (Messiah, 2014), statistical physics (Wannier, 1987), solid-state physics (Grosso & Parravicini, 2013), and many-body theory (Inkson, 2012), posing significant challenges due to its complexity, inter-disciplinarity, data-scarcity, and demand for precise mathematical formulation evaluation.

To address these challenges and test the performance of LLMs in modern physical science, we present CMPhysBench, a novel benchmark specifically designed to evaluate the problem-solving abilities of LLMs in CMP. It comprises 520 questions, manually authored by Ph.D. students and postdoctoral researchers based on standard graduate textbooks spanning key CMP subfields, with difficulty levels ranging from undergraduate to advanced graduate coursework. Unlike multiple-choice benchmarks (Saikh et al., 2022; Yue et al., 2025) that are ignorant of intermediate reasoning and procedural correctness, CMPhysBench emphasizes open-ended calculation problems, requiring models to produce complete solutions that reflect both conceptual understanding and computational precision. Furthermore, to quantify the differences between mathematical responses and handle

various answer types, we propose the Scalable Expression Edit Distance (SEED) metric shown in Figure 1. The SEED metric is inspired by Expression Edit Distance (EED) (Qiu et al., 2025b) and offers a more robust and interpretable performance measure than exact string matching (He et al., 2024) .

To summarize, our contribution lies in the following aspects:

- **Graduate-level CMP benchmark with open-ended calculation.** We release *CMPhys-Bench*, a 520-question benchmark *manually authored by Ph.D. students and postdoctoral researchers* based on standard graduate textbooks, spanning core subfields and emphasizing open-ended calculation tasks that require complete, step-by-step solutions across five answer types.
- **SEED: fine-grained, accurate evaluation metric.** We propose the *Scalable Expression Edit Distance* (SEED), which maps diverse answer types to ASTs and computes tree-edit distance with unit conversion, scientific-notation parsing, and rounding within tolerance, yielding non-binary partial credit and interpretable error localization.
- **Comprehensive empirical study and diagnosis.** We evaluate *18* proprietary and open-source LLMs on CMPhysBench, finding consistently low performance and pronounced variability across models, and providing quantitative analyses that illuminate failure modes and opportunities for improving domain-specific reasoning in CMP. Our experimental also results reveal a notable performance gap between mathematical reasoning and physical reasoning in CMP.

## 2 CMPHYSBENCH

The whole CMPhysBench benchmark consists of three parts: dataset overview, data curation and evaluation metric.

### 2.1 OVERVIEW

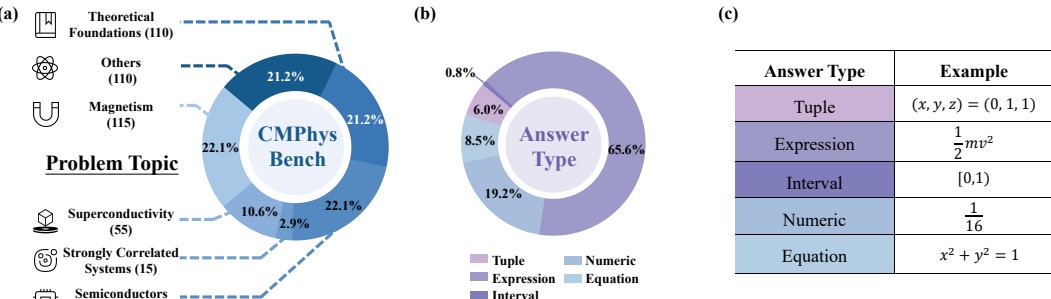

Figure 2: Overview of the CMPhysBench dataset and answer types. (a) Distribution of problem topics across various condensed matter physics domains in CMPhysBench. (b) Distribution of answer types across the dataset, highlighting the prevalence of numeric answers. (c) A table displaying examples of each answer type.

As shown in Table 1 in Appendix B , CMPhysBench covers 520 carefully curated questions with difficulty spanning from introductory undergraduate exercises to advanced graduate-level challenges from CMP. CMPhysBench comprises six representative topics of CMP, structured as follows. Firstly, to ensure domain representativeness, we include four core topics: *Magnetism*, *Superconductivity*, *Strongly Correlated Systems*, and *Semiconductors*. Secondly, to holistically evaluate LLMs beyond narrow domain expertise, we extend the benchmark with two additional dimensions of CMP. One of the additional categories is *Theoretical Foundations*, which encompasses, crystallography, plasmonics, phase transitions, and condensed matter field theory. The other is *Others*, which further includes quantum mechanics, statistical physics, electrodynamics, and quantum field theory. This *hierarchical categorization* allows simultaneous assessment of domain-specific knowledge and general physical reasoning capabilities.

At the same time, following the settings in OlympiadBench (He et al., 2024), we also categorize these questions based on different answer types. Specifically, there are five answer types in CMPhysBench, including tuple, equation, numeric, expression, and interval. The categorization of the questions is performed by human experts to ensure its correctness. Details of the data categorization and distribution are listed in Figure 2(a) and (b), and our benchmark contains topics across various fields in condensed-matter physics, and the problems can be divided into five types: Tuple, Numeric, Expression, Equation and Interval, and the examples of them are shown in Figure. 2 (c).

## 2.2 DATA CURATION

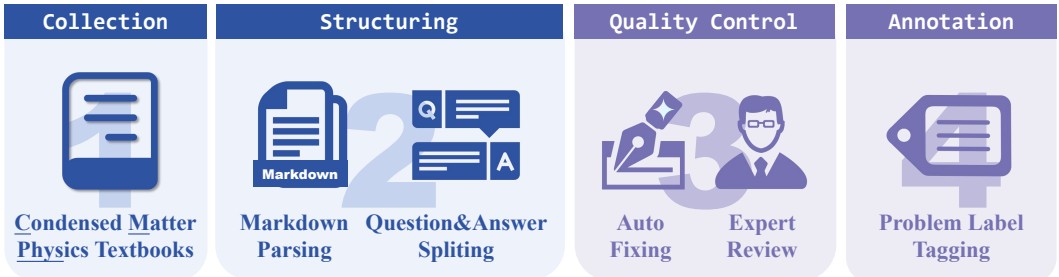

Figure 3: The data curation pipeline of CMPhysBench.

We initially collect course materials and exercise problems from 17 textbooks with difficulty spanning from introductory undergraduate exercises to advanced graduate-level challenges. We mainly choose classical textbooks in CMP like *An Introduction to Quantum Field Theory* (Peskin, 2018), *Classical Field Theory* (Soper, 2008), *Condensed Matter Field Theory (3rd edition)* (Altland & Simons, 2010), *Introduction to Many-Body Physics* (Coleman, 2015), *Statistical Physics* (Landau & Lifshitz, 1980) etc. As shown in Figure 3, the data curation pipeline consists of four stages to ensure the quality and usability of the benchmark.

**Collection.** Firstly, the collected textbook materials are first converted from PDF to Markdown format, followed by a transformation into structured, machine-readable text formats. Specifically, we convert the PDF documents of textbooks into Markdown format via MathPix[1].

**Structuring.** Subsequently, we carefully modify the selected the problems relevant to calculation tasks and adapted them to a standardized calculation-question format suitable for benchmarking. Specifically, we propose only calculation problems.

**Quality Control, Expert Review and Annotation.** Finally, each adapted question is *manually checked by Ph.D. students and postdoctoral researchers specialized in Condensed Matter Physics*. During this review process, incomprehensible or ambiguous questions are removed and detailed answers and solutions were carefully verified, ensuring that all retained data could be clearly interpreted and evaluated. In addition, all questions are further classified based on the type of answer they require, demonstrated by Figure 2 (c).

## 2.3 EVALUATION METRIC: SCALABLE EXPRESSION EDIT DISTANCE (SEED)

To provide a robust and fine-grained evaluation, we follow the core EED pipeline. We first extract the mathematical expression from the model output and canonicalize it to standard LaTeX. we then convert it to a SymPy[2] object via `latex2sympy_extended`, normalize terms to a positive canonical form, and apply `simplify()` to stabilize and accelerate subsequent comparison.

While EED struggles with noisy LaTeX and varied answer types, SEED standardizes them and provides fine-grained, physics-aware evaluation. We extend the evaluation in three directions. First, *answer-type support and unification* (as shown in right side of Figure 4): (1) Expressions are directly parsed into abstract syntax trees (ASTs). (2) Equations are standardized by moving all terms to one side. (3) Tuples are evaluated component-wise by positional matching, and the SEED scores

---

[1]`https://mathpix.com/`
[2]`https://www.sympy.org/`

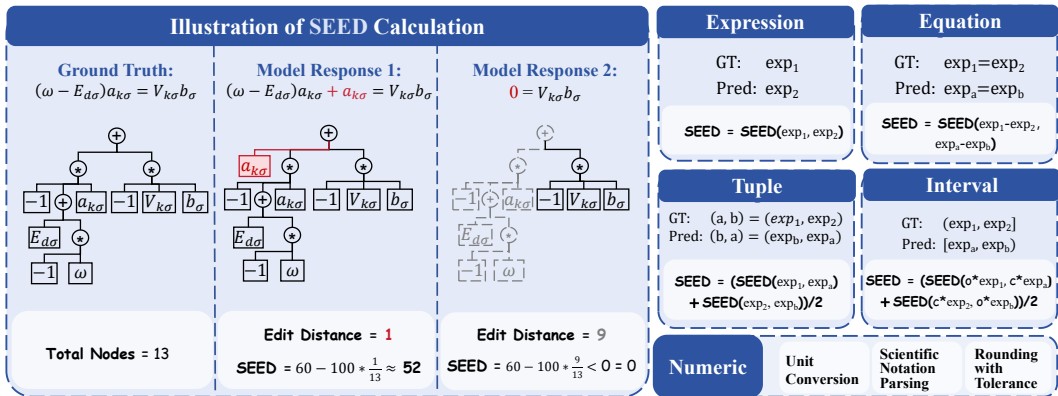

Figure 4: SEED calculation process for different answer types, including edit-distance examples and rules for expressions, equations, tuples, intervals, and numeric answers. For a detailed explanation of the SEED scoring function, see the Appendix C.

are averaged. (4) Intervals incorporate boundary openness through symbolic representations. (5) Numeric answers are evaluated with attention to unit conversion, scientific notation parsing, and rounding within relative tolerance. Second, *expanded symbolic coverage*: we add native handling of matrices/vectors and inequalities ($<, \leq, >, \geq$), which we canonicalize as $f(\cdot) \# 0$ (with $\# \in \{<, \leq, >, \geq\}$) while preserving semantics under operations that flip inequality direction. Third, *robust LaTeX preprocessing*: we strip wrappers such as \boxed{}, remove \left and \right, normalize implicit multiplication (e.g., $2x, ab$), unify Unicode symbols (e.g., the minus sign), standardize function aliases and font commands (\mathrm{}, \mathcal{}, \mathbb{}), discard extraneous natural-language boilerplate (e.g., "Final Answer:"), and auto-balance parentheses and fractions. These improvements enable SEED to build ASTs reliably from noisy LLM outputs and, via tree-edit distance, deliver non-binary partial credit together with interpretable error localization.

Its type-agnostic AST design and pluggable, physics-aware normalization allow easy extension to new answer types and domain rules, enabling application across CMP and other STEM tasks while maintaining unified, fine-grained evaluation.

## 3 EXPERIMENTS

### 3.1 MODELS

We group models by provider families: *OpenAI* (GPT-4o (OpenAI, 2024a); o1 (OpenAI, 2024b); o3 (OpenAI, 2025b); o3-mini (OpenAI, 2025a); o4-mini (OpenAI, 2025b)), *Google* (Gemini 2.5 Pro, Gemini 2.0 Flash Thinking (Team et al., 2023)), *Anthropic* (Claude 3.7 Sonnet; Claude 3.7 Sonnet Thinking (Anthropic, 2025)), *xAI* (Grok 3 Beta (AI, 2025), Grok 4), *Meta/Llama* (Llama-3.1-70B-Instruct; Llama-3.3-70B-Instruct (Grattafiori et al., 2024)), *Alibaba/Qwen* (Qwen3-32B (Team, 2025a); QWQ-32B (Team, 2025b)), and *DeepSeek* (DeepSeek-V3 (Deepseek, 2024); DeepSeek-R1 and its distilled variants—R1-Distill-Llama-70B, R1-Distill-Qwen-32B (Guo et al., 2025)). This family-based taxonomy spans both proprietary and open-source ecosystems as well as general-purpose and Long-CoT reasoning models, enabling controlled comparisons on CMPhysBench.

### 3.2 EXPERIMENT SETUP

For proprietary LLMs, we utilize API services to query these models. Meanwhile, for DeepSeek-v3 and DeepSeek-R1, due to their requirement on huge GPU memory, we also adopt API services for the query. In contrast, for the remaining open-source general and reasoning LLMs, we adopt vllm[3] for parallel acceleration.

---

[3] https://docs.vllm.ai/

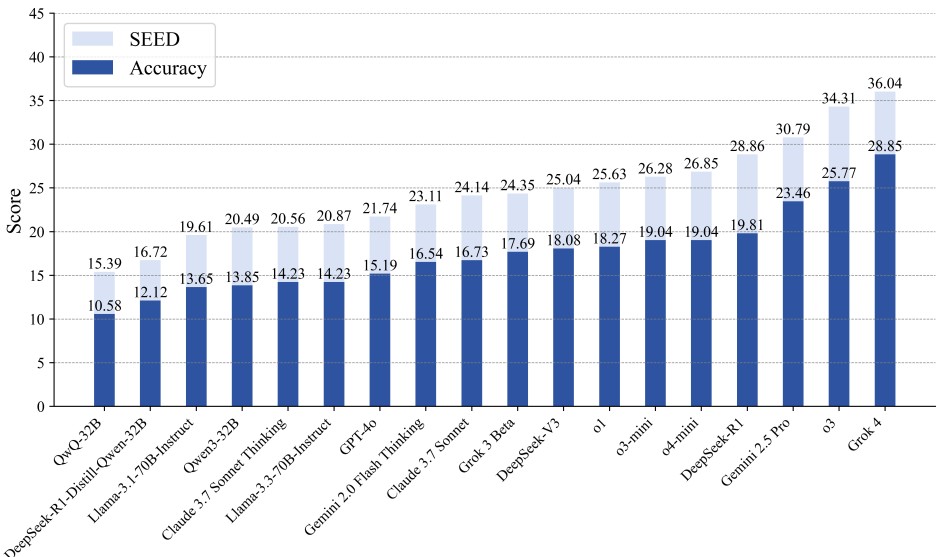

Figure 5: Model performance on CMPhysBench. For each model, we report the SEED score along with the expert-labeled accuracy.

## 3.3 MAIN RESULTS

As shown in Figure 5, CMPhysBench is challenging across the board. A small lead cluster, Grok 4, o3, and Gemini 2.5 Pro, achieves scores ranging from 30 to 36 on the SEED score scale, with expert-labeled accuracies between 23% and 29% (e.g., Grok 4 achieves 36.0 SEED score and 28.9% accuracy). This cluster clearly separates from the mid pack. Most remaining systems lie in a middle band (approximately 23–28 SEED score; 16–20% accuracy), while instruction-tuned open-source baselines fall lower (20–22 SEED; 14–15% accuracy), and distilled/smaller variants are the weakest (15–17 SEED score; 10–12% accuracy).

However, an interesting phenomenon suggests that reasoning LLMs may not perform better than general LLMs on these challenging domain-specific problems in condensed matter physics, because the problems require domain-specific knowledge and become highly difficult, making it easy for reasoning models to make mistakes during the reasoning process, which then will propagate to the final answer. In this case, the more LLMs think, the more likely they could make a mistake. We also observe many near-miss solutions (e.g., unit handling, constants, boundary conditions): expert-labeled accuracy is strict and stays low, whereas SEED systematically yields higher values (typically +5–9 points) by explicitly crediting partial correctness. Collectively, these patterns provide a more comprehensive understanding of prevailing limitations of LLMs and underscore the necessity of physics-aware training and evaluation protocols.

## 4 DISCUSSION

### 4.1 ERROR ANALYSIS

*LLMs can make many types of mistakes.* To investigate model failure patterns on CMPhysBench, we conduct a detailed error analysis by passing incorrect predictions to GPT-4o and prompting it to infer the underlying reasons. To ensure the reliability of this automated approach, inspired by recent work like xVerify, we validated it against a set of 300 diverse question-response pairs manually annotated by domain experts. Our method achieved a 98% agreement rate with human consensus, giving us high confidence in its ability to serve as a scalable proxy for expert evaluation. This allows us to categorize error types in a consistent manner. Notably, Grok 4 is excluded from this analysis as it does not generate intermediate reasoning chains, making it difficult to assess its internal logic. Based

on an initial classification by domain experts, errors are grouped into eight categories, as detailed in Figure 6.

Figure 6: Analysis of error types across models. Left: Error count breakdown by type for each model on CMPhysBench. Right: Representative examples for each error type, where the background color corresponds to the error category in the left plot. Blue text highlights the specific error location, and the reason is provided below each example.

As shown in Figure 6 and Table 4 in the appendix, the following two errors account for a significant proportion: Concept and Model Misuse and Mathematical or Logical Errors. Concept and Model Misuse are the most dominant error type, and account for over 40–50% of all normalized errors in models such as GPT-4o (66.5%), Claude 3.7 Sonnet Thinking (51.6%), and DeepSeek-V3 (56.3%). This indicates that many models, even high-performing ones, struggle with the correct application of domain-specific physical principles. Another major category is Mathematical or Logical Errors, typically contributing 20–30% of total errors. For instance, o4-mini and o3 exhibit logical mistake rates of 31.0% and 29.4%, respectively, despite having relatively good task-following ability. These issues range from incorrect algebraic manipulation to invalid approximations and reveal persistent gaps in symbolic reasoning.

Task Misunderstanding is more prominent in instruction-tuned models like Qwen3-32B (24.2%) and QwQ-32B (27.0%), which often fail to interpret specific constraints. In contrast, more advanced models such as Gemini 2.5 Pro and o3 demonstrate better prompt adherence, with lower task misunderstanding rates (e.g., Gemini 2.5 Pro: 7.5%), suggesting that superior reasoning techniques improve problem comprehension. While other error types like Unit Errors remain rare ($<2\%$), the overall analysis underscores the need for improved scientific alignment and symbolic precision. This diagnostic analysis provides a direct roadmap for mitigating these failures, with specific improvement directions detailed in Appendix G. Furthermore, it allows us to form concrete hypotheses about future, domain-specific models. We hypothesize that fine-tuning will reduce knowledge-based "Concept and Model Misuse" errors, while core "Mathematical or Logical Errors" may persist. A key goal of CMPhysBench is to provide a platform to test such hypotheses.

## 4.2 ANALYSIS OF DIFFERENT PROBLEM TOPICS

As shown in Figure 7(a) and Table 5 in the appendix, performance varies markedly across topics and model families. Grok 4 leads most categories, achieving the highest scores in Magnetism (35.30), Superconductivity (43.42), and Theory (41.21). Meanwhile, o3 demonstrates strong all-around performance, placing first in Others (46.42) and second in Superconductivity (35.77), Strongly Correlated Systems (37.34), and Semiconductors (27.80). Topic-specific peaks also emerge: DeepSeek-R1 attains the best score in Strongly Correlated Systems (42.16), Gemini 2.5 Pro leads in Semiconductors (29.18) and is competitive in Theory (40.50), and DeepSeek-V3 ranks second in Magnetism (25.75). Notably, even top models display pronounced asymmetries; for example, Grok 4 is strong in Superconductivity and Theory but weaker in Strongly Correlated Systems, indicating that strengths do not transfer uniformly across CMP subfields.

These patterns highlight the importance of domain-specific reasoning over generic mathematical skill. Although instruction-tuned open-source baselines generally trail proprietary models, some exhibit

localized strengths. For instance, Qwen3-32B performs relatively well in Theory with a score of 35.47 but remains weak in Magnetism (8.47), underscoring its uneven competence across topics. This cross-domain spread suggests the need for subfield-aware training.

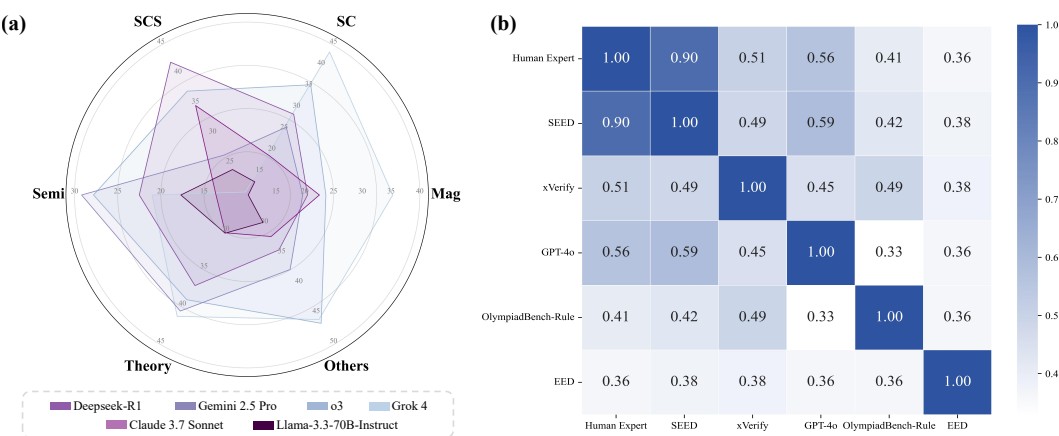

Figure 7: Comparison of model performance and metric correlations. (a) Radar chart of model performance across six domains. Abbreviations: **Mag** = Magnetism, **SC** = Superconductivity, **SCS** = Strongly Correlated Systems, **Semi** = Semiconductors, **Theory** = Theoretical Foundations, **Others** = Others. (b) Spearman correlation between human expert ratings and automatic evaluation metrics.

### 4.3 COMPARISON WITH DIFFERENT METRICS

To systematically assess the reliability and alignment of various evaluation metrics, we compare SEED against four widely used alternatives: Expression Edit Distance (EED) (Qiu et al., 2025b), GPT-4o-based judgment (OpenAI, 2024a), xVerify-9B-C (Chen et al., 2025), OlympiadBench-rule based metric (He et al., 2024) and human labels. Human experts have labeled answers as strictly correct (1) or incorrect (0). We then converted the SEED score into a corresponding binary value, where only a perfect score (SEED = 100) was considered correct (1). Spearman correlation coefficients between these metrics and human expert ratings are shown in Figure 7(b). *SEED exhibits the highest correlation with human experts ($\rho = 0.90$)*, demonstrating superior agreement with expert judgment. This performance stems from SEED's design as a discrete, structure-aware metric that supports partial credit and accommodates a wide range of symbolic answer types commonly found in CMP, such as equations, intervals, and tuples. Unlike binary accuracy metrics, SEED distinguishes near-miss cases from completely incorrect outputs, providing a more nuanced assessment of symbolic reasoning. Furthermore, SEED is designed for polynomial expression similarity evaluation which is very common in graduate-level CMP.

In contrast, *EED*, though fast and interpretable, struggles with generalization beyond simple expressions. It fails to handle complex structures like equations with symbolic manipulations or multi-component answers. *GPT-4o* and *xVerify*, while more flexible in language understanding, are less reliable for evaluating highly structured mathematical expressions. Their performance ($\rho = 0.56$ and $0.51$, respectively) suggests limitations in symbolic alignment, particularly for multi-step derivations and dense expressions common in CMP problems. Specifically, these two evaluation methods do not explicitly consider equivalent transformation of expression, making it not be the most suitable metric in CMP. *OlympiadBench-Rule* supports multiple answer types, but its rule-based approach is overly simplistic and often fails to account for meaningful structural or mathematical equivalence, resulting in the lowest correlation ($\rho = 0.41$).

To summarize, these findings indicate that SEED provides *fine-grained partial correctness credit and robustness, alongside wide applicability and interpretability*, making it a stronger metric for domain-specific scientific reasoning.

## 5 RELATED WORK

Due to the rapid development of LLMs and their potential in scientific research, there is a growing trend toward evaluating their performance on scientific problems. For example, benchmarks such as SciQ (Welbl et al., 2017), ScienceQA (Saikh et al., 2022), ARC (Clark et al., 2018), OpenBookQA (Mihaylov et al., 2018), PubMedQA (Jin et al., 2019), SciBench (Wang et al., 2024), SciEval (Sun et al., 2023), and E-Eval (Hou et al., 2024) provide platforms for testing LLMs on general scientific questions across multiple disciplines. Normally, these benchmarks cover a broad spectrum of topics but often cap difficulty at K-12 or introductory college levels and favor multiple-choice formats, which increasingly lag behind frontier models and limit exploration of deeper scientific reasoning, especially in physics. In contrast, emerging benchmarks like UGPhysics (Xu et al., 2025a), GPQA (Rein et al., 2024), SuperGPQA (Du et al., 2025), PHYSICS (Zheng et al., 2025), SciCode (Tian et al., 2024), PHYBench (Qiu et al., 2025b), and PhysReason (Zhang et al., 2025b) raise the bar by introducing undergraduate- to graduate-level problems, step- or expression-aware grading, and physics-specific evaluation pipelines, which impose stricter requirements on domain knowledge, reasoning, and problem-solving. However, most of these still emphasize broad coverage rather than depth within a specific research direction; they do not thoroughly examine sustained knowledge acquisition and structured derivations in narrowly defined subfields. In summary, while existing work has substantially advanced the evaluation of LLMs' physics problem-solving abilities, there remains a notable gap for benchmarks that probe rigorous, subfield-specific physics tasks with fine-grained, structure-aware scoring.

## 6 CONCLUSION

In this work, we have introduce CMPhysBench, a novel benchmark tailored to evaluate the proficiency of LLMs in the domain of Condensed Matter Physics. Comprising 520 carefully selected questions based on authoritative textbooks, CMPhysBench encompasses a wide range of representative topics such as magnetism, superconductivity, strongly correlated systems, semiconductors, etc. To ensure accurate evaluation, we propose the Scalable Expression Edit Distance (SEED) score to measure the similarity between various mathematical expressions. Our findings reveal a significant performance gap, with LLMs excelling in general mathematical tasks yet falling short in the specialized context of Condensed Matter Physics, which further underscores the necessity to enhance the effectiveness of LLMs in this domain. Further, we believe domain-specific dataset is crucial in promoting the performance of LLM in the future.

## ACKNOWLEDGEMENT

We thank the anonymous reviewers and area chair for their helpful comments. This work was supported by Shanghai Artificial Intelligence Laboratory, Shanghai Municipal Science and Technology Major Project (Grant No. 2025SHZDZX025D06), New Generation Artificial Intelligence-National Science and Technology Major Project (Grant No. 2025ZD0121802), Intern-Discovery and the Postdoctoral Fellowship Program of CPSF (Grant No. GZC20232943).

## ETHICS STATEMENT

We study the performance of LLMs in condensed matter physics, and this work does not involve explicit ethic issues.

## REPRODICIBILITY STATEMENT

Our SEED score calculation code and CMPhysBench benchmark questions are placed in the supplementary to ensure the reproducibility of the article.

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

## A  OVERVIEW OF THE APPENDIX

Section B contains details about the composition of CMPhysBench, the data curation process, and comparisons with existing benchmarks, highlighting its uniqueness and advantages in the domain of condensed matter physics.

Section C introduces the SEED evaluation metric and compares it against EED, EM, and GPT-4o-based scoring, demonstrating SEED's scalability and improved alignment with human judgment in symbolic reasoning tasks. It also contains metrics for evaluating complex reasoning related work.

Section D outlines the experimental settings, including prompt design, tested models, and implementation details used for both answer generation and error analysis.

Section E presents an in-depth analysis of model performance on CMPhysBench, including breakdowns by error type and topics, as well as representative case studies of physics problems and model predictions.

Section F discloses our use of Large Language Models in the preparation of this manuscript.

Section G discusses future research directions suggested by our error analysis. This includes domain-specific fine-tuning to address conceptual misuse, neuro-symbolic methods for mathematical errors, and instruction tuning for system limitations, in addition to leveraging the SEED metric for advanced training paradigms.

Section H provides a detailed explanation of the notations used in Figure 1, covering fundamental operators, creation and annihilation operators, energy parameters, and the Green's function, to aid readers unfamiliar with quantum many-body physics.

## B  CMPHYSBENCH DETAILS

### B.1  COMPOSITION OF CMPHYSBENCH

In this study, we categorize the benchmark question set into six major domains: **Magnetism**, **Superconductivity**, **Strongly Correlated Systems**, **Semiconductors**, **Theoretical Foundations** and **General Concepts**, as shown in Figure 2. Each domain encompasses key theoretical frameworks and representative problems appropriate for graduate-level physics education, reflecting a progressive trajectory from foundational understanding to advanced modeling.

- **Theoretical Foundations** encompass a wide range of topics from quantum field theory (e.g., Klein-Gordon fields, Dirac fields, path integrals, spontaneous symmetry breaking) to statistical physics (e.g., Gibbs distribution, fluctuation theory). Given their *central role in supporting advanced topics and their broad applicability*, this domain also includes 110 questions, aiming to reinforce a systematic understanding of modern theoretical physics.
- **Magnetism** and **Semiconductors** are each represented by 115 questions. These domains focus on phenomena such as spin dynamics, magnetic interactions, charge transport, band theory, and device-level behavior—topics of both fundamental and applied significance in condensed matter physics and materials science. The higher question volume reflects *the practical complexity and frequency of these systems in real-world physical problems*, encouraging students to develop robust modeling and analytical skills.
- **Superconductivity** includes topics such as the macroscopic Ginzburg–Landau theory, microscopic BCS theory, and related experimental phenomena. Although conceptually challenging, the theory is relatively self-contained and often revolves around paradigmatic problems. Thus, a moderate number of questions (55) is sufficient to assess students' *depth of understanding through carefully selected*, representative examples.
- **Strongly Correlated Systems** cover advanced topics such as quantum many-body fluctuations, the Hubbard model, and Mott transitions. As one of the most intellectually demanding and research-intensive areas in theoretical physics, it is included as an extension module with 15 high-level questions. These problems are designed to *challenge students with strong theoretical backgrounds and facilitate further exploration of frontier topics*.
- **Others** cover fundamental problems and computational techniques in quantum mechanics, including harmonic oscillators, perturbation theory, and spin systems. As these topics span multiple

subfields and *serve as essential tools across the curriculum*, a relatively large number of questions (110) are assigned to this domain to ensure comprehensive training in basic problem-solving skills and physical intuition.

Generally, the distribution of questions reflects both the structural organization of knowledge in graduate-level physics and a deliberate balance between representativeness, theoretical depth, computational rigor, and pedagogical utility. The design seeks to ensure both breadth and depth, enabling the benchmark to serve as a comprehensive tool for assessing general competence while also identifying advanced reasoning capabilities.

Furthermore, following the settings in OlympiadBench (He et al., 2024), we also categorize these questions based on the answer types. Specifically, there are five answer types in CMPhysBench, including tuple, equation, numeric, expression, and interval, whose distributions are illustrated in Figure 2. The categorization of the questions is performed by human experts to ensure its correctness.

## B.2 COMPARISON WITH OTHER BENCHMARKS

Table 1: Comparison of our benchmark with existing datasets. For Level: COMP = Competition level, CEE = University Entrance Exam, K1–K12 = Primary and Secondary School. For Question Type: OE = Open-ended, MC = Multiple-choice.

| Benchmark | Size | Level | Question Type | Scoring Type |
|---|---|---|---|---|
| JEEBench (Arora et al., 2023) | 123 | CEE | OE, MC | Binary |
| GPQA (Rein et al., 2024) | 227 | Graduate | OE | Binary |
| SciQ (Welbl et al., 2017) | 13,679 | K4–K8 | OE, MC | Binary |
| SciEval (Sun et al., 2023) | 1,657 | — | OE, MC | Binary |
| SciBench (Wang et al., 2024) | 295 | University | OE | Binary |
| ScienceQA (Saikh et al., 2022) | 617 | K1–K12 | MC | Binary |
| MMMU (Yue et al., 2024) | 443 | University | OE, MC | Binary |
| MMMU-Pro (Yue et al., 2025) | 3,460 | University | MC | Binary |
| OlympiadBench (He et al., 2024) | 2,334 | COMP | OE | Binary |
| EMMA (Hao et al., 2025) | 156 | — | MC | Binary |
| PHYSICS (Feng et al., 2025) | 1,297 | University | OE | Binary |
| SciCode (Tian et al., 2024) | 338 | University | OE | Binary |
| PhySense (Xu et al., 2025b) | 380 | University-Graduate | OE, MC | Binary |
| PHYBench (Qiu et al., 2025a) | 500 | K10–COMP | OE | Detailed |
| **CMPhysBench** | **520** | **Graduate** | **OE** | **Detailed** |

Table 1 provides a comparison between CMPhysBench and a range of existing scientific and physics-related benchmarks. While earlier benchmarks such as PHYSICS, PHYBench, and SciBench have advanced the development of AI systems capable of handling domain-specific problems, CMPhysBench distinguishes itself through its graduate-level difficulty, richer answer representations, and more robust evaluation protocol.

Unlike PHYBench, where open-ended (OE) questions are limited to symbolic expressions and evaluated using EED (Expression Edit Distance), CMPhysBench introduces a more powerful and extensible metric named SEED (Scalable Expression Edit Distance). This allows for nuanced grading and flexible equivalence matching beyond symbolic forms.

Key distinctions of CMPhysBench include:

- **Advanced Answer Types**: Answers are not restricted to expressions or numerics; they also include tuples, intervals, and equation systems, reflecting the diversity of physical reasoning and solution strategies required in real-world scientific practice.
- **Graduate-Level Scope**: Questions are curated from advanced textbooks and course materials in theoretical and condensed matter physics, ensuring alignment with the cognitive demands of graduate education and early-stage research, rather than standard undergraduate or competition-level problems.

- **Semantic Evaluation Flexibility**: The SEED metric enables fine-grained evaluation that supports partial credit, symbolic and numeric equivalence, and structural matching—offering more meaningful feedback on models' reasoning capabilities.

In contrast, many prior benchmarks (e.g., PHYSICS, MMMU, ScienceQA) focus on multiple-choice formats or expression-only open-ended questions at the high school or early undergraduate level, and often rely on binary correctness. CMPhysBench, by contrast, aims to bridge the gap between academic problem-solving and scientific reasoning, providing a more rigorous, diverse, and research-oriented benchmark for evaluating LLMsin physics and beyond.

## C  EVALUATION METRIC

### C.1  SCALABLE EXPRESSION EDIT DISTANCE

| Feature | Original EED | Our SEED Method |
|---------|-------------|-----------------|
| Supported Structures | Simple Expressions | Expressions, Equations, Tuples, Intervals |
| Parse Tree Nodes | Basic (symbols/functions) | Extended (Matrices, Derivatives, Inequalities) |
| Preprocessing | Minimal | Extensive Standardization and Disambiguation |
| Robustness | Limited | Enhanced Parsing Robustness |

Table 2: Comparison of SEED and original EED.

In this part, we briefly introduce the differences and advantages of our proposed **Scalable Expression Edit Distance (SEED)** compared with the original Expression Edit Distance (EED). The term "scalable" refers to our method's capability of extending to more complex and varied answer types, including intervals, tuples, and equations, beyond the simple mathematical expressions handled by EED. Key differences and advantages are listed as follows.

1. **Enhanced Expression Parsing:**
   SEED supports parsing and scoring of complex LaTeX structures including matrices, derivative expressions (e.g., $\frac{d}{dx}$), logical relations ($=, <, >$), and various special formatting cases, significantly extending EED's capabilities.
2. **Extended Node Types in Parse Trees:**
   Beyond basic numeric, constant, and symbolic nodes, SEED introduces dedicated nodes for matrices, inequalities, derivatives, and logical operators, ensuring richer semantic representations.
3. **Advanced Preprocessing and Standardization:**
   SEED standardizes special fonts (e.g., `\mathscr{L}`), derivative notations, exponent formats, vector notations, fraction formats, and removes problematic LaTeX commands (e.g., `\text{}`), significantly reducing parsing ambiguities and errors.
4. **Support for Varied Answer Types:**
   - **Expressions:** Handled similarly to EED, with improved robustness and accuracy.
   - **Equations:** SEED extracts both sides of equations separately and then combines them into a unified form (typically by subtraction) for scoring. This approach allows direct handling of equation-type answers, addressing EED's inability to process equations effectively.
   - **Tuples:** Answers structured as tuples (e.g., $(a, b, c) = (1, 2, 3)$) are transformed into key-value pairs, allowing structured and accurate component-wise evaluation.
   - **Intervals:** Interval expressions (e.g., $(a, b)$) are transformed into evaluable mathematical forms, including explicit handling of open and closed boundaries, to facilitate robust scoring.
5. **Robust Symbol and Format Handling:**
   Enhanced recognition logic prevents parsing errors from similar LaTeX commands (e.g., distinguishing `\left` from `\le`), and uniformly standardizes ambiguous formatting and special characters.

Beyond these structural improvements, SEED provides a fine-grained, non-binary score by quantifying the similarity between the predicted and ground-truth expression trees. The score is calculated based on the relative edit distance, $r$, between the ground-truth tree ($T_{gt}$) and the generated tree ($T_{gen}$). This scoring function is adapted from the methodology used in PHYBench (Qiu et al., 2025b) and is

defined as follows:

$$r = \frac{\text{Distance}(T_{\text{gt}}, T_{\text{gen}})}{\text{Size}(T_{\text{gt}})}, \quad \text{score} = \begin{cases} 100, & \text{if } r = 0 \text{ (exact match)}, \\ 60 - 100r, & 0 < r < 0.6, \\ 0, & r > 0.6. \end{cases}$$

Here, Distance$(\cdot, \cdot)$ is the tree-edit distance and Size$(\cdot)$ is the number of nodes in the tree. This function assigns a full score of 100 for a perfect match, linearly scales the score down from a baseline of 60 to award partial credit for answers with minor errors, and assigns a score of 0 for expressions that are significantly incorrect ($r \geq 0.6$).

## C.2 RELATED WORK: METRICS FOR EVALUATING COMPLEX REASONING

The evaluation of complex reasoning in artificial intelligence, a critical aspect of measuring progress in the field, has evolved significantly beyond simple accuracy metrics. As models become more sophisticated, so too must the methods we use to assess their capabilities, moving towards more nuanced and comprehensive techniques. Evaluation methods for complex reasoning broadly fall into four families. (1) *Outcome-based scoring*. Many benchmarks judge only the final answer via exact match (EM), e.g., GSM8K (Cobbe et al., 2021) and MATH (Hendrycks et al., 2021), sometimes with minor normalization, which is simple but brittle to equivalent forms and formatting noise. To reduce false negatives, several pipelines (Lewkowycz et al., 2022; Hendrycks et al., 2021) augment EM with CAS-based checks using SymPy to test symbolic/numeric equivalence (and lightweight tolerances), as popularized by Minerva and now embedded in common evaluators. Recent math (Karki et al., 2025) suites further combine exact, numerical, and symbolic equivalence in a single grader. (2) *Fine-grained structure-aware similarity*. Instead of only the final token string, expression-level metrics compare the structure of predicted and reference solutions. PHYBench's Expression Edit Distance (Qiu et al., 2025b) computes tree-edit distances over SymPy expression trees and converts them to a fine-grained score, capturing "almost-correct" derivations that EM misses. (3) *Judge- and verifier-based evaluation*. LLM-as-a-Judge (Gu et al., 2024; Chen et al., 2024) offers flexible rubric-style grading but is susceptible to systematic biases (e.g., position/verbosity), motivating protocols and debiasing to improve reliability. In contrast, lightweight answer verifiers target objective tasks by extracting the final answer from long chains and checking equivalence across formats; recent models such as xVerify (Chen et al., 2025) report strong accuracy across math/short-answer settings. Toolkits like MARIO-Eval (Zhang et al., 2024) unify CAS checks with optional LLM judging to improve robustness across datasets. Overall, recent trends move from brittle EM toward type-aware, fine-grained structure-aware, and process-aware evaluation, often blending CAS equivalence, expression-level distances, and calibrated judges/verifiers to better match expert judgments on complex reasoning.

# D EXPERIMENTAL DETAILS

## D.1 PROMPTS FOR RESPONSE GENERATION

This prompt is designed to assess a model's ability to perform symbolic, step-by-step reasoning in advanced physics. The model must use only the symbols provided, avoiding any external assumptions, and present the final result in a clear LaTeX `\\boxed{}` format. This ensures precision, interpretability, and alignment with expert-level problem-solving.

---

***Prompts for Response Generation***

You are a condensed matter physics expert. Please read the following question and provide a step-by-step solution using only the given symbols. Do not introduce any new symbols that are not provided in the problem statement. Your final answer must be presented as a readable LaTeX formula, enclosed in a `\\boxed{}` environment.

---

### D.2 PROMPTS FOR ERROR ANALYSIS

This prompt instructs GPT-4o, acting as a physics expert, to systematically evaluate model-generated answers by checking correctness, categorizing errors (e.g., conceptual, mathematical, dimensional) and providing concise reasoning. Responses are structured in JSON format, enabling precise and efficient error analysis and scoring.

---

**Prompts for Error Analysis**

You are a condensed matter physics expert. Your task is to evaluate a model-generated answer to a physics question.
Please perform the following:
1. Determine whether the model's answer is correct.
2. If incorrect, identify which of the following error categories (a–h) the answer falls into (multiple selections allowed):
a) Concept and Model Misuse: Misuse or misapplication of core physical principles, laws, or models (e.g., using Newtonian mechanics in relativistic regimes).
b) Task Misunderstanding: Misunderstanding of what the question is asking (e.g., solving for the wrong quantity, or ignoring critical constraints).
c) Mathematical or Logical Errors: Incorrect mathematical manipulations, derivations, or reasoning steps (e.g., algebraic mistakes, sign errors, invalid inferences).
d) Notational Inconsistency: Incorrect, inconsistent, or ambiguous use of symbols or notation (e.g., mixing variables, wrong subscripts, undefined terms).
e) Unit or Dimensional Errors: Violations of dimensional consistency or incorrect unit conversions (e.g., adding quantities of different dimensions).
f) Approximation Misuse: Applying approximations or assumptions that are unjustified in the given context (e.g., small-angle approximation where angle is large).
g) System Limitations: Errors clearly stemming from generation failures, hallucinations, or limitations of the AI system (e.g., nonsensical steps, abrupt output truncation).
h) Redundant or Irrelevant Content: Inclusion of content that is redundant, off-topic, or distracts from the solution (e.g., repeating known facts or copying question text unnecessarily).
Respond in JSON format as follows:
{ "is_correct": "true" or "false", "error_types": ["a", "c", ...], "explanation": "Your reasoning in 1–2 sentences" }
Question: {question}
Ground Truth: {ground_truth}
Model Response: {model_response}

---

## D.3 MODELS AND SETTINGS

We evaluate a diverse set of proprietary and open-source large language models, as summarized in Table 3. For OpenAI (GPT-4o, o1, o3, o4-mini) and Anthropic (Claude 3 series) models etc, we use their official APIs. Google Gemini and xAI Grok models are also accessed via respective APIs. For open-source models such as Qwen, DeepSeek, and LLaMA variants, we employ the `vLLM` inference engine for efficient batched decoding. In cases where `vLLM` is not supported (e.g., vision-language models), we fall back to the HuggingFace `transformers` library for direct model loading.

| Model | Param | Src | URL |
|---|---|---|---|
| QwQ-32B | temperature = 0.6 | local checkpoint | `https://huggingface.co/Qwen/QwQ-32B` |
| DeepSeek-R1-Distill-Qwen-32B | temperature = 0.6 | local checkpoint | `https://huggingface.co/deepseek-ai/DeepSeek-R1-Distill-Qwen-32B` |
| Qwen3-32B | temperature = 0.6 | local checkpoint | `https://huggingface.co/Qwen/Qwen3-32B` |
| DeepSeek-R1-Distill-Llama-70B | temperature = 0.6 | local checkpoint | `https://huggingface.co/deepseek-ai/DeepSeek-R1-Distill-Llama-70B` |
| Llama-3.1-70B-Instruct | temperature = 0.6 | local checkpoint | `https://huggingface.co/meta-llama/Llama-3.1-70B-Instruct` |
| Llama-3.3-70B-Instruct | temperature = 0.6 | local checkpoint | `https://huggingface.co/meta-llama/Llama-3.3-70B-Instruct` |
| Claude-3-7-Sonnet | - | claude-3-7-sonnet-latest | `https://www.anthropic.com/` |
| Claude-3-7-Sonnet-thinking | - | claude-3-7-sonnet-thinking | `https://www.anthropic.com/` |
| GPT-4o | - | OpenAI | `https://platform.openai.com` |
| o1 | - | o1 | `https://platform.openai.com` |
| o3-mini | - | o3-mini | `https://platform.openai.com` |
| o3 | - | o3 | `https://platform.openai.com` |
| o4-mini | - | o4-mini | `https://platform.openai.com` |
| DeepSeek-R1 | - | deepseek-r1 | `https://huggingface.co/deepseek-ai/DeepSeek-R1` |
| DeepSeek-V3 | - | deepseek-v3 | `https://huggingface.co/deepseek-ai/DeepSeek-V3` |
| Gemini-2.0-flash-thinking | - | gemini-2.0-flash-thinking-exp | `https://ai.google.dev/` |
| Gemini-2.5-pro | - | gemini-2.5-pro-preview-03-25 | `https://ai.google.dev/` |
| Grok-3-Beta | - | grok-3-beta | `https://x.ai/` |
| Grok-4 | - | grok-4 | `https://x.ai/` |

Table 3: The sources of models used in the experiments and the hyperparameters configuration. "-" stands for default parameters.

# E  EXPERIMENT RESULTS

## E.1  ERROR TYPES COUNTS

Table 4: Error types counts. Abbreviations: **CM** = Concept and Model Misuse, **ML** = Mathematical or Logical Errors, **UD** = Unit or Dimensional Errors, **TM** = Task Misunderstanding, **SL** = System Limitations, **NI** = Notational Inconsistency, **RI** = Redundant or Irrelevant Content, **AM** = Approximation Misuse.

| Model | CM | ML | UD | TM | SL | NI | RI | AM |
|---|---|---|---|---|---|---|---|---|
| QwQ-32B | 202 | 64 | 0 | 64 | 123 | 2 | 10 | 0 |
| DeepSeek-R1-Distill-Qwen-32B | 281 | 91 | 3 | 34 | 43 | 3 | 2 | 0 |
| Llama-3.1-70B-Instruct | 314 | 102 | 2 | 21 | 2 | 3 | 3 | 2 |
| Qwen3-32B | 233 | 72 | 0 | 64 | 73 | 0 | 5 | 1 |
| Claude 3.7 Sonnet Thinking | 131 | 104 | 0 | 21 | 184 | 1 | 4 | 1 |
| Llama-3.3-70B-Instruct | 317 | 97 | 3 | 20 | 4 | 3 | 1 | 1 |
| GPT-4o | 305 | 114 | 4 | 12 | 4 | 2 | 0 | 0 |
| Gemini 2.0 Flash Thinking | 269 | 118 | 2 | 24 | 8 | 8 | 5 | 0 |
| Claude 3.7 Sonnet | 257 | 150 | 7 | 11 | 0 | 8 | 0 | 0 |
| Grok 3 Beta | 263 | 105 | 4 | 44 | 3 | 4 | 3 | 2 |
| DeepSeek-V3 | 272 | 130 | 3 | 11 | 0 | 8 | 1 | 1 |
| o1 | 266 | 114 | 7 | 19 | 4 | 10 | 5 | 0 |
| o3-mini | 250 | 117 | 5 | 26 | 1 | 13 | 9 | 0 |
| o4-mini | 235 | 128 | 5 | 28 | 1 | 15 | 9 | 0 |
| DeepSeek-R1 | 247 | 124 | 1 | 20 | 13 | 10 | 2 | 0 |
| Gemini 2.5 Pro | 215 | 132 | 4 | 30 | 4 | 9 | 4 | 0 |
| o3 | 191 | 151 | 5 | 20 | 0 | 12 | 4 | 3 |

## E.2 MODEL PERFORMANCE ON DIFFERENT DOMAINS

Table 5: Model performance across condensed matter physics domains (normalized scores, two decimal places). Abbreviations: **All** = SEED of all problems, **Mag** = Magnetism, **SC** = Superconductivity, **SCS** = Strongly Correlated Systems, **Semi** = Semiconductors, **Theory** = Theoretical Foundations, **Others** = Others.   Blue   = highest,   Purple   = second highest in each column.

| Model | All | Mag | SC | SCS | Semi | Theory | Others |
|---|---|---|---|---|---|---|---|
| QwQ-32B | 15.39 | 8.93 | 8.75 | 26.29 | 14.97 | 22.23 | 17.56 |
| DeepSeek-R1-Distill-Qwen-32B | 16.72 | 8.41 | 12.65 | 20.12 | 12.30 | 24.01 | 24.31 |
| Llama-3.1-70B-Instruct | 19.61 | 8.56 | 9.30 | 29.63 | 19.05 | 27.24 | 27.92 |
| Qwen3-32B | 20.49 | 8.47 | 15.65 | 17.25 | 16.30 | 35.47 | 25.32 |
| Claude 3.7 Sonnet Thinking | 20.56 | 10.68 | 22.38 | 24.53 | 13.65 | 33.44 | 23.77 |
| Llama-3.3-70B-Instruct | 20.87 | 10.19 | 13.08 | 24.25 | 17.68 | 30.10 | 29.58 |
| GPT-4o | 21.74 | 19.04 | 18.90 | 29.03 | 11.58 | 28.95 | 28.42 |
| Gemini 2.0 Flash Thinking | 23.11 | 13.85 | 13.47 | 11.15 | 26.66 | 29.82 | 28.85 |
| Claude 3.7 Sonnet | 24.14 | 22.55 | 19.13 | 34.93 | 13.61 | 30.05 | 31.93 |
| Grok 3 Beta | 24.35 | 17.74 | 26.34 | 26.74 | 16.26 | 34.37 | 28.39 |
| DeepSeek-V3 | 25.04 | 25.75 | 29.67 | 9.25 | 15.30 | 27.73 | 31.62 |
| o1 | 25.63 | 23.75 | 26.02 | 28.42 | 12.72 | 32.39 | 33.78 |
| o3-mini | 26.28 | 19.51 | 27.67 | 19.08 | 14.40 | 35.63 | 36.72 |
| o4-mini | 26.85 | 17.50 | 27.63 | 22.32 | 18.33 | 38.13 | 34.49 |
| DeepSeek-R1 | 28.86 | 20.49 | 28.88 | 42.16 | 22.50 | 37.10 | 34.18 |
| Gemini 2.5 Pro | 30.79 | 19.65 | 25.86 | 26.67 | 29.18 | 40.50 | 37.44 |
| o3 | 34.31 | 23.68 | 35.77 | 37.34 | 27.80 | 38.99 | 46.42 |
| Grok 4 | 36.04 | 35.30 | 43.42 | 20.49 | 21.01 | 41.21 | 45.79 |

## E.3 CMPHYSBENCH PROBLEM AND RESPONSES DEMO

---

**51: Interacting Fields and Feynman Diagrams, Decay of a scalar particle**

**Problem:** Decay of a scalar particle. Consider the following Lagrangian, involving two real scalar fields $\Phi$ and $\phi$ :

$$\mathcal{L} = \frac{1}{2}(\partial_\mu \Phi)^2 - \frac{1}{2}M^2\Phi^2 + \frac{1}{2}(\partial_\mu \phi)^2 - \frac{1}{2}m^2\phi^2 - \mu\Phi\phi\phi.$$

The last term is an interaction that allows a $\Phi$ particle to decay into two $\phi$ 's, provided that $M > 2m$. Assuming that this condition is met, calculate the lifetime of the $\Phi$ to lowest order in $\mu$.

**Final Answer:**

$$\tau = \frac{8\pi M}{\mu^2}(1 - \frac{4m^2}{M^2})^{-1/2}$$

---

**Gemini-2.5-pro Answer:**
The lifetime of a particle is the inverse of its total decay rate $\Gamma$. The decay process is $\Phi \to \phi + \phi$. We will calculate $\Gamma$ to the lowest order in the coupling constant $\mu$.
**Step 1: Determine the matrix element $\mathcal{M}$.**
The Lagrangian density is given by:

$$\mathcal{L} = \frac{1}{2}(\partial_\mu \Phi)^2 - \frac{1}{2}M^2\Phi^2 + \frac{1}{2}(\partial_\mu \phi)^2 - \frac{1}{2}m^2\phi^2 - \mu\Phi\phi\phi.$$

The interaction term responsible for the decay is $\mathcal{L}_{int} = -\mu\Phi\phi\phi$. This is a trilinear interaction. The Feynman rule for the vertex connecting one $\Phi$ particle and two $\phi$ particles is derived from this term. Since the two $\phi$ fields are identical, there is a symmetry factor of $2! = 2$ associated with permutations of these fields if they were externally labelled, or equivalently, one considers the derivative of the interaction term with respect to the fields. For an interaction term of the form $-\frac{g_{abc}}{N!}\Phi_a\Phi_b\Phi_c$ where $N$ is the number of identical fields of a certain type, the vertex factor is $-ig_{abc}$. Our interaction is $-\mu\Phi\phi^2$. This can be written as $-\frac{2\mu}{2!}\Phi\phi^2$. Thus, the vertex factor is $-i(2\mu)$. The matrix element $\mathcal{M}$ for the decay $\Phi \to \phi + \phi$ is therefore $2\mu$ (by convention, the $-i$

---

is often factored out). So, $|\mathcal{M}|^2 = (2\mu)^2 = 4\mu^2$.

**Step 2: Write down the formula for the decay rate $\Gamma$.**
For a particle of mass $M$ decaying from its rest frame into two final state particles with momenta $p_1$ and $p_2$, the decay rate is given by:

$$\Gamma = \frac{1}{2M} \int \frac{d^3p_1}{(2\pi)^3 2E_1} \int \frac{d^3p_2}{(2\pi)^3 2E_2} (2\pi)^4 \delta^{(4)}(P - p_1 - p_2)|\mathcal{M}|^2 S$$

Here, $P = (M, \vec{0})$ is the four-momentum of the initial $\Phi$ particle. $p_1 = (E_1, \vec{p}_1)$ and $p_2 = (E_2, \vec{p}_2)$ are the four-momenta of the two outgoing $\phi$ particles. $E_1 = \sqrt{|\vec{p}_1|^2 + m^2}$ and $E_2 = \sqrt{|\vec{p}_2|^2 + m^2}$. Since the two final state $\phi$ particles are identical, the statistical factor $S = \frac{1}{2!} = \frac{1}{2}$.
Substituting $|\mathcal{M}|^2 = 4\mu^2$ and $S = 1/2$:

$$\Gamma = \frac{1}{2M} \cdot \frac{1}{2} \cdot (4\mu^2) \int \frac{d^3p_1}{(2\pi)^3 2E_1} \int \frac{d^3p_2}{(2\pi)^3 2E_2} (2\pi)^4 \delta^{(4)}(P - p_1 - p_2)$$

$$\Gamma = \frac{\mu^2}{M} \int \frac{d^3p_1}{(2\pi)^3 2E_1} \int \frac{d^3p_2}{(2\pi)^3 2E_2} (2\pi)^4 \delta(M - E_1 - E_2)\delta^{(3)}(\vec{0} - \vec{p}_1 - \vec{p}_2)$$

**Step 3: Evaluate the phase space integral.**
Let $I$ be the phase space integral:

$$I = \int \frac{d^3p_1}{(2\pi)^3 2E_1} \int \frac{d^3p_2}{(2\pi)^3 2E_2} (2\pi)^4 \delta(M - E_1 - E_2)\delta^{(3)}(\vec{p}_1 + \vec{p}_2)$$

Integrate over $\vec{p}_2$ using the $\delta^{(3)}(\vec{p}_1 + \vec{p}_2)$, which sets $\vec{p}_2 = -\vec{p}_1$. This implies $|\vec{p}_2| = |\vec{p}_1|$. Let $p_f = |\vec{p}_1|$. Then $E_1 = E_2 = E_f = \sqrt{p_f^2 + m^2}$.

$$I = \int \frac{d^3p_1}{(2\pi)^6 (2E_f)(2E_f)} (2\pi)^4 \delta(M - 2E_f) = \frac{1}{(2\pi)^2} \int \frac{d^3p_1}{4E_f^2} \delta(M - 2E_f)$$

Convert $d^3p_1$ to spherical coordinates: $d^3p_1 = p_f^2 dp_f d\Omega$. The angular integration gives $4\pi$ as the integrand is spherically symmetric.

$$I = \frac{4\pi}{(2\pi)^2} \int_0^\infty \frac{p_f^2 dp_f}{4E_f^2} \delta(M - 2E_f) = \frac{1}{4\pi} \int_0^\infty \frac{p_f^2 dp_f}{E_f^2} \delta(M - 2E_f)$$

The delta function $\delta(M - 2E_f)$ constrains $E_f = M/2$. Since $E_f = \sqrt{p_f^2 + m^2}$, we have $M/2 = \sqrt{p_f^2 + m^2}$, which gives $M^2/4 = p_f^2 + m^2$. So, $p_f^2 = M^2/4 - m^2$. For $p_f$ to be real, $M^2/4 > m^2$, i.e., $M > 2m$, which is given. The magnitude of the final momentum is $p_f = \sqrt{M^2/4 - m^2} = \frac{1}{2}\sqrt{M^2 - 4m^2}$. To evaluate the integral over $dp_f$, we use the property $\int g(x)\delta(f(x))dx = \sum_i \frac{g(x_i)}{|f'(x_i)|}$ where $f(x_i) = 0$. Here $x = p_f$, $g(p_f) = p_f^2/E_f^2$, and $f(p_f) = M - 2E_f = M - 2\sqrt{p_f^2 + m^2}$. $f'(p_f) = -2\frac{dE_f}{dp_f} = -2\frac{p_f}{E_f}$. At $E_f = M/2$, $p_f = p_0 = \frac{1}{2}\sqrt{M^2 - 4m^2}$. $|f'(p_0)| = \frac{2p_0}{M/2} = \frac{4p_0}{M}$. So the integral is $\frac{p_0^2/E_0^2}{|f'(p_0)|} = \frac{p_0^2}{(M/2)^2} \frac{M}{4p_0} = \frac{4p_0^2}{M^2} \frac{M}{4p_0} = \frac{p_0}{M}$. Substituting this into the expression for $I$:

$$I = \frac{1}{4\pi} \frac{p_0}{M} = \frac{1}{4\pi M} \frac{\sqrt{M^2 - 4m^2}}{2} = \frac{\sqrt{M^2 - 4m^2}}{8\pi M}$$

This is a standard result for the two-body phase space factor.

**Step 4: Combine results to get $\Gamma$.**
Substitute the value of $I$ back into the expression for $\Gamma$:

$$\Gamma = \frac{\mu^2}{M} I = \frac{\mu^2}{M} \frac{\sqrt{M^2 - 4m^2}}{8\pi M} = \frac{\mu^2 \sqrt{M^2 - 4m^2}}{8\pi M^2}$$

**Step 5: Calculate the lifetime $\tau$.**
The lifetime $\tau$ is the inverse of the decay rate $\Gamma$:

$$\tau = \frac{1}{\Gamma} = \frac{8\pi M^2}{\mu^2 \sqrt{M^2 - 4m^2}}$$

The condition $M > 2m$ ensures that the term under the square root is positive, so the decay is kinematically allowed and the lifetime is real and positive. The calculation is to the lowest order in $\mu$, which means considering only the tree-level diagram, resulting in $\Gamma \propto \mu^2$.
Final check of units: In natural units ($\hbar = c = 1$), mass $M, m$ and coupling $\mu$ have units of energy $[E]$. The Lagrangian density $\mathcal{L}$ has units $[E]^4$. The kinetic terms $(\partial_\mu \Phi)^2$ imply $\Phi$ has units $[E]$. The interaction term $\mu \Phi \phi \phi$ has units $[\mu][E][E][E]$. So $[\mu]$ must be $[E]$. The decay rate $\Gamma$ has units of $[E]$ (inverse time). Our result for $\Gamma$:
$\frac{[\mu]^2 \sqrt{[M]^2 - [m]^2}}{[M]^2} = \frac{[E]^2 \sqrt{[E]^2}}{[E]^2} = \frac{[E]^2 [E]}{[E]^2} = [E]$. This is correct. The lifetime $\tau$ has units of $[E]^{-1}$ (time). Our result for $\tau$: $\frac{[M]^2}{[\mu]^2 \sqrt{[M]^2 - [m]^2}} = \frac{[E]^2}{[E]^2 [E]} = [E]^{-1}$. This is correct.

The final expression for the lifetime $\tau$ is:

$$\boxed{\tau = \frac{8\pi M^2}{\mu^2 \sqrt{M^2 - 4m^2}}}$$

**SEED score:**
100

**o3 Answer:**
Tree-level (lowest order in the coupling
$mu$)
**1. Vertex rule**
From the interaction term $-\mu \, \Phi \phi \phi$ the (constant) Feynman amplitude for $\Phi \to \phi \phi$ is $-\mathrm{i}\,\mu$ , so that

$$|\mathcal{M}|^2 = \mu^2$$

.

**2. Two–body phase–space factor**
For a particle of mass $M$ decaying at rest into two identical particles of mass $m$, the Lorentz–invariant phase space gives

$$\int \mathrm{dPS}_2 = \frac{1}{8\pi} \sqrt{1 - \frac{4m^2}{M^2}} \; .$$

Because the two final particles are identical, an extra factor $1/2!$ is required.

**3. Decay probability per unit time**

$$\left(\text{probability per unit time}\right) = \frac{1}{2M} \left(\tfrac{1}{2!}\right) |\mathcal{M}|^2 \int \mathrm{dPS}_2 = \frac{\mu^2}{32\pi M} \sqrt{1 - \frac{4m^2}{M^2}} \; .$$

**4. Lifetime** The lifetime is the inverse of the above quantity, hence

$$\boxed{\text{lifetime} = \frac{32\pi \, M}{\mu^2 \sqrt{1 - \dfrac{4m^2}{M^2}}}}$$

which is valid provided $M > 2m$.

**SEED score:**
77.5

**GPT-4o Comments:**
The model answer contains a mathematical error in calculating the decay rate: it shows a factor of 32 instead of the correct factor of 8 present in the reference answer. The error likely arises from incorrect consideration of symmetry or phase space factors.

**Expert Comments:**
The analysis is correct. The model wrongly calculate the symmetry factor, and this dates back to the problem. The error type can be logical error or concept misunderstanding. In another word, the model know partially about the physical concept.

**Error Type:**
"c": Mathematical or Logical Errors

---

**DeepSeek-R1 Answer:**
The lifetime of the $\Phi$ particle decaying into two $\Phi$ particles is calculated using the decay rate $\Gamma$, which is determined by the interaction term in the Lagrangian. The decay rate to lowest order in $\mu$ is given by:

$$\Gamma = \frac{\mu^2 \sqrt{M^2 - 4m^2}}{32\pi M^2}$$

The lifetime $\tau$ is the inverse of the decay rate:

$$\tau = \frac{1}{\Gamma} = \frac{32\pi M^2}{\mu^2 \sqrt{M^2 - 4m^2}}$$

**Final Answer:**

$$\boxed{\tau = \frac{32\pi M^2}{\mu^2 \sqrt{M^2 - 4m^2}}}$$

**SEED score:**
77.5

**Comments:**
The model's answer is mathematically incorrect. When the ground truth expression is simplified, it becomes $\tau = \dfrac{8\pi M^2}{\mu^2 \sqrt{M^2 - 4m^2}}$, which differs from the model's answer by a factor of 4 in the numerical coefficient (32 in the model vs. 8 in the ground truth).

**Expert Comments:**
The analysis is correct. The model wrongly calculate the symmetry factor, and this dates back to the problem. The error type can be logical error. In another word, the model know partially about the physical concept because it calculate the right target but give a wrong answer.

**Error Type:**
Mathematical or Logical Errors

---

**DeepSeek-V3 Answer:**
To calculate the lifetime of the $\Phi$ particle decaying into two $\phi$ particles, we'll follow these steps:

1. **Identify the Interaction Term**:
The interaction term in the Lagrangian is $-\mu\Phi\phi\phi$, which allows the decay $\Phi \to \phi\phi$.

2. **Compute the Decay Amplitude**:
The Feynman rule for the vertex $\Phi\phi\phi$ is $-i\mu$ (since the interaction term is $-\mu\Phi\phi\phi$, the vertex factor is $-i\mu$).

3. **Phase Space Calculation**:
The decay rate $\Gamma$ for $\Phi \to \phi\phi$ in the rest frame of $\Phi$ is given by:

$$\Gamma = \frac{1}{2M} \int \frac{d^3 p_1}{(2\pi)^3 2E_1} \frac{d^3 p_2}{(2\pi)^3 2E_2} (2\pi)^4 \delta^4(P - p_1 - p_2)|\mathcal{M}|^2,$$

where $|\mathcal{M}|^2$ is the squared matrix element for the decay.

**Matrix Element Squared**:

For the decay $\Phi \to \phi\phi$, the matrix element is $\mathcal{M} = -i\mu$. Thus, $|\mathcal{M}|^2 = \mu^2$.

**Phase Space Integral**:

The phase space integral for a two-body decay with identical particles (symmetry factor $1/2!$) gives:

$$\Gamma = \frac{1}{2M}\left(\frac{1}{2}\right)\int \frac{d^3p_1 d^3p_2}{(2\pi)^2 4E_1 E_2}(2\pi)^4\delta(M - E_1 - E_2)\delta^3(\mathbf{p}_1 + \mathbf{p}_2)\mu^2.$$

Integrating over $\mathbf{p}_2$ and using $\mathbf{p}_1 = -\mathbf{p}_2$, we have:

$$\Gamma = \frac{\mu^2}{16\pi M}\int \frac{d^3p_1}{E_1^2}\delta(M - 2E_1).$$

Using $E_1 = \sqrt{m^2 + |\mathbf{p}_1|^2}$, and $d^3p_1 = 4\pi|\mathbf{p}_1|^2 d|\mathbf{p}_1|$, the delta function enforces:

$$2E_1 = M \implies |\mathbf{p}_1| = \frac{1}{2}\sqrt{M^2 - 4m^2}.$$

Evaluating the integral:

$$\Gamma = \frac{\mu^2}{16\pi M}\left(4\pi\frac{|\mathbf{p}_1|}{4E_1^2}\right)\Bigg|_{E_1 = M/2}.$$

Since $E_1 = M/2$ and $|\mathbf{p}_1| = \frac{1}{2}\sqrt{M^2 - 4m^2}$, we get:

$$\Gamma = \frac{\mu^2}{16\pi M}\left(\frac{\pi}{M}\sqrt{1 - \frac{4m^2}{M^2}}\right).$$

4. **Final Decay Rate**:

Simplifying, the decay rate is:

$$\Gamma = \frac{\mu^2}{16\pi M}\sqrt{1 - \frac{4m^2}{M^2}}.$$

5. **Lifetime Calculation**:

The lifetime $\tau$ of the $\Phi$ particle is the inverse of the decay rate:

$$\tau = \frac{1}{\Gamma} = \frac{16\pi M}{\mu^2\sqrt{1 - \frac{4m^2}{M^2}}}.$$

The final result for the lifetime of the $\Phi$ particle is:

$$\boxed{\tau = \frac{16\pi M}{\mu^2\sqrt{1 - \frac{4m^2}{M^2}}}}$$

**SEED score:**
55

**Comments:**
The model's answer for the lifetime $\tau$ is twice the correct value. The correct expression is $\tau = \frac{8\pi M}{\mu^2}(1 - \frac{4m^2}{M^2})^{-1/2}$. The model's answer has a numerator of $16\pi M$ instead of $8\pi M$. This discrepancy by a factor of 2 could arise from a conceptual misunderstanding (e.g., incorrect matrix element $|\mathcal{M}|^2$ or symmetry factor) or a mathematical error in the phase space integration.

**Expert Comments:**
The analysis is partially correct, and model knows partially about the concept but wrongly calculate the integral.

**Error Type:**
Concept and Model Misuse, Mathematical or Logical Errors

**Grok-4-0709 Answer:**

$$\frac{8\pi M^2}{\mu^2 \sqrt{M^2 - 4m^2}}$$

**SEED score:**
100

---

### 228: Movement of electrons in a crystal in electric and magnetic fields

**Problem:** N atoms form a two-dimensional square lattice, with each atom contributing one electron to form a two-dimensional free electron gas. The electron energy expression is

$$E(k) = \frac{\hbar^2 k_x^2}{2m} + \frac{\hbar^2 k_y^2}{2m}$$

Derive the formula for the density of states of a two-dimensional free gas.
**Final Answer:**

$$g(E) = \frac{m}{\pi \hbar^2}$$

---

**DeepSeek-R1 Answer:**
To derive the density of states (DOS) formula for a two-dimensional free electron gas, we first consider an annulus in k-space corresponding to iso-energy contours. The energy expression is:

$$E = \frac{\hbar^2 (k_x^2 + k_y^2)}{2m} = \frac{\hbar^2 k^2}{2m},$$

where $k = \sqrt{k_x^2 + k_y^2}$. In k-space, the iso-energy contour is a circle of radius $k$. The area of the annulus corresponding to energies from $E$ to $E + dE$ (i.e., between radii $k$ and $k + dk$) is $2\pi k\, dk$.
The number of allowed k-states per unit area of k-space is $(L/(2\pi))^2$, where $L$ is the linear dimension of the system (and $A = L^2$ is the real-space area). The number of states $dN$ within the k-space annulus $2\pi k\, dk$, per unit real-space area, is:

$$\frac{dN}{A} = \frac{1}{L^2} \cdot \left(\frac{L}{2\pi}\right)^2 \cdot 2\pi k\, dk = \frac{k\, dk}{2\pi}.$$

From $E = \hbar^2 k^2/(2m)$, differentiation yields $dk = \frac{m}{\hbar^2 k} dE$. Substituting this into the expression for the number of states per unit real-space area, we obtain the density of states $g(E) = \frac{dN}{A \cdot dE}$:

$$g(E) = \frac{1}{dE}\left(\frac{k\, dk}{2\pi}\right)$$
$$= \frac{k}{2\pi}\frac{dk}{dE}$$
$$= \frac{k}{2\pi} \cdot \frac{m}{\hbar^2 k}$$
$$= \frac{m}{2\pi \hbar^2}.$$

The final result for the density of states is:

$$g(E) = \frac{m}{2\pi \hbar^2}$$

**SEED score:**
49

**Comments:**
The model's answer for the 2D density of states, $g(E) = \dfrac{m}{2\pi\hbar^2}$, is incorrect because it is missing the spin degeneracy factor of 2, or rather, it has an extra factor of 1/2. The correct density of states for a 2D free electron gas, including spin, is $g(E) = \dfrac{m}{\pi\hbar^2}$.

**Expert Comments:**
The analysis is correct. The model do not understand the concept of spin and this leads to the ignorance of spin degeneracy factor 2.

**Error Type:**
Concept and Model Misuse

**DeepSeek-V3 Answer:**
To derive the density of states $g(E)$ for a two-dimensional free electron gas, follow the steps below:

**1. Energy dispersion relation**:
The energy of an electron is given by

$$E(k) = \frac{\hbar^2(k_x^2 + k_y^2)}{2m} = \frac{\hbar^2 k^2}{2m}$$

where $k = \sqrt{k_x^2 + k_y^2}$.

**2. Number of states in wavevector space**:
In two-dimensional $k$-space, the area per state is $(2\pi/L)^2$, where $L$ is the side length of the 2D lattice. For a system composed of $N$ atoms, the total $k$-space area is determined by a circle with radius from 0 to $k_F$ ($k_F$ is the Fermi wavevector). The number of states is

$$\text{Number of states} = \left(\frac{L}{2\pi}\right)^2 \cdot \pi k^2$$

because the area in $k$-space is $\pi k^2$.

**3. Relation between total number of electrons and energy**:
Each state can accommodate 2 electrons (spin up and down), so the total number of electrons $N$ satisfies:

$$N = 2 \cdot \left(\frac{L}{2\pi}\right)^2 \cdot \pi k_F^2$$

From this, we find the Fermi wavevector $k_F$ as

$$k_F^2 = \frac{2\pi N}{L^2}$$

or $k_F = \sqrt{2\pi n}$, where $n = N/L^2$ is the areal electron density.

**4. Definition of density of states**:
The density of states $g(E)$ is the number of states per unit energy interval (including spin degeneracy), i.e.,

$$g(E) = \frac{dN}{dE}$$

From steps 2 and 3, we obtain the relation between number of states and energy $E$:

$$N(E) = 2 \cdot \left(\frac{L}{2\pi}\right)^2 \cdot \pi k^2$$

Substituting $k^2 = \frac{2mE}{\hbar^2}$, we get

$$N(E) = 2 \cdot \left(\frac{L}{2\pi}\right)^2 \cdot \pi \left(\frac{2mE}{\hbar^2}\right) = \frac{mL^2}{\pi\hbar^2} E$$

**5. Differentiate to obtain the density of states**:
Differentiating $N(E)$ gives $g(E)$:

$$g(E) = \frac{dN}{dE} = \frac{mL^2}{\pi\hbar^2}$$

This is the total density of states of the system. The density of states per unit area is:

$$g(E) = \frac{g(E)}{L^2} = \frac{m}{\pi\hbar^2}$$

**6. Final result**:
The density of states for a two-dimensional free electron gas is a constant, independent of energy $E$:

$$\boxed{g(E) = \frac{m}{\pi\hbar^2}}$$

**SEED score:**
100

**Grok-4-0709 Answer:**
The density of states per unit area without magnetic field is $g(E)$.
In the presence of magnetic field $B$, the energy levels condense into Landau levels with energy spacing $\Delta E$.
The number of states per unit area in an energy interval $\Delta E$ without magnetic field is $g(E)\Delta E$.
These states condense into one Landau level, so the degeneracy per unit area is $D = g(E)\Delta E$.

$$\boxed{D = g(E)\Delta E}$$

**SEED score:**
0

**Comments:** The model's answer incorrectly applies the concept of degeneracy, as it relies on the density of states and energy interval, without addressing how the Landau level degeneracy is determined by the magnetic field. The correct expression for Landau level degeneracy depends on magnetic flux through the lattice area, yielding the given formula:$D = \frac{eB}{\pi\hbar}$.

**Expert Comments:**
The analysis is correct. The model do not understand both the problem and the concept, especially the Landau level because this problem do not contain a magnetic field, leading to a wrong answer.

**Error Type:**
Concept and Model Misuse

---

**116: Electrostatics of Dielectrics, Thermodynamic relations for dielectrics in an electric field**

**Problem:**
Determine the height $h$ by which the liquid surface inside a vertical parallel-plate capacitor rises.
**Final Answer:**

$$h = \frac{\varepsilon - 1}{8\pi\rho g}E^2$$

**gemini-2.0-flash-thinking Answer:**
Here's a step-by-step solution to determine the height $h$ by which the liquid rises inside the capacitor, using only the provided symbols:
**1. Energy considerations:**
The free energy change, $\tilde{F}$, must account for the gravitational potential energy increase due to the liquid rising and the reduction in electrostatic energy due to the introduction of the dielectric.

**2. Gravitational Potential Energy:**

The increase in gravitational potential energy per unit area of the capacitor plates is $\rho g h(h/2) = \frac{1}{2}\rho g h^2$.

**3. Electrostatic Energy:**

The electric field is $E$ without the liquid. When the liquid of dielectric constant $\varepsilon$ is present, it displaces air (dielectric constant = 1). The energy density reduction is proportional to $(\varepsilon - 1)E^2$. Integrating this energy density reduction over the volume the liquid occupies, we get the reduction in electrostatic energy per unit area of the capacitor plates as $\frac{1}{2}(\varepsilon - 1)\epsilon_0 E^2 h$. However, we can express this as $\frac{1}{2}(\varepsilon - 1)DEh$ where $D = \epsilon_0 E$ is the electric displacement. Furthermore, because the field is held constant here using a voltage source, we have $\frac{1}{2}(\varepsilon-1)\epsilon_0 E^2 h$

**4. Free Energy Minimization:**

The system will adjust $h$ to minimize the free energy $\tilde{F}$. This means the change in free energy due to a small change in $h$ must be zero, i.e., $\frac{\partial \tilde{F}}{\partial h} = 0$. Therefore, we are balancing the gain in gravitational potential energy with the loss of electrostatic energy.

**5. Balancing Energies:**

We set the change in gravitational potential energy equal to the \*negative\* of the change in electrostatic energy (per unit area) when the liquid rises to height $h$:
$\frac{1}{2}\rho g h^2 = \frac{1}{2}(\varepsilon - 1)\epsilon_0 E^2 h$ . However, $\epsilon_0$ is not in our permitted variables so we have to remove it. Since $E$ is constant, and the parallel plate capacitor can be expressed as $E = V/d$, where $d$ is the separation between the plates and $V$ is the voltage. The capacitance is $C = \epsilon A/d$. So $\epsilon$ represents relative permittivity which is $\varepsilon = \frac{\epsilon}{\epsilon_0}$ so $\epsilon_0 = \frac{\epsilon}{\varepsilon}$.
Then we can get $\frac{1}{2}\rho g h^2 = \frac{1}{2}(\varepsilon - 1)\epsilon_0 E^2 h$, which becomes
$\frac{1}{2}\rho g h^2 = \frac{1}{2}(\varepsilon - 1)\frac{\epsilon}{\varepsilon}E^2 h$

**6. Solving for h:**

We solve for $h$ by dividing both sides by $h/2$, we have:

$$\rho g h = (\varepsilon - 1)\frac{\epsilon}{\varepsilon}E^2$$

$$h = \frac{(\varepsilon - 1)E^2\epsilon}{\rho g \varepsilon}$$

**7. Final Expression:**

$$\boxed{h = \frac{(\varepsilon - 1)E^2\epsilon}{\rho g \varepsilon}}$$

**SEED score:**
36.5

**Comments:**
The model's derivation is fundamentally incorrect as it starts with dimensionally inconsistent and physically flawed expressions for both gravitational potential energy and electrostatic energy. It also demonstrates severe notational confusion, using the symbol '$\epsilon$' for both relative and absolute permittivity, which culminates in a nonsensical final equation.

**Expert Comments:**
The analysis is correct, and the units required in the problem is misunderstood by the model.

**Error Type:**
Concept and Model Misuse

---

**Gemini-2.5-pro Answer:**

Here is a step-by-step derivation to determine the height $h$.

The liquid rises to a height $h$ where the total free energy of the system, $\tilde{F}$, is at a minimum. This equilibrium condition is found by setting the derivative of the total free energy with respect to the height $h$ to zero. The total free energy is a sum of the electrical free energy and the gravitational potential energy.

**Step 1: Gravitational Potential Energy**

As a volume of liquid rises into the capacitor, its gravitational potential energy increases. Let's consider the cross-sectional area of the capacitor gap to be $A_{cs}$. The volume of the raised liquid is $V = A_{cs}h$. The mass of this liquid is $m = \rho V = \rho A_{cs}h$. The center of mass of this raised liquid column is at a height of $h/2$. The increase in gravitational potential energy, $U_g$, is therefore:
$U_g = mg\frac{h}{2} = (\rho A_{cs}h)g\frac{h}{2} = \frac{1}{2}\rho g A_{cs}h^2$

**Step 2: Electrical Free Energy**

The problem specifies that $\tilde{F}$ is the relevant free energy, which is minimized when the potential across the plates is held constant. When the liquid with dielectric constant $\varepsilon$ rises, it displaces the air (with $\varepsilon_{air} \approx 1$). This changes the electrical energy stored in the capacitor.

The problem's use of the symbol $\pi$ and the absence of the permittivity of free space $\varepsilon_0$ indicates that Gaussian units are appropriate. In these units, the energy density of an electric field in a linear dielectric is $u = \frac{\varepsilon}{8\pi}E^2$.

The energy density in the air-filled region is $u_{air} = \frac{1}{8\pi}E^2$. The energy density in the liquid-filled region is $u_{liquid} = \frac{\varepsilon}{8\pi}E^2$.

When the liquid rises by a height $h$, a volume $V = A_{cs}h$ previously filled with air is now filled with the liquid. The change in the stored electrical energy, $\Delta U_{elec}$, in this volume is:
$\Delta U_{elec} = (u_{liquid} - u_{air})V = \left(\frac{\varepsilon}{8\pi}E^2 - \frac{1}{8\pi}E^2\right)A_{cs}h = \frac{\varepsilon-1}{8\pi}E^2 A_{cs}h$

For a system at constant potential, the change in the relevant free energy, $\Delta \tilde{F}_{elec}$, is the negative of the change in the stored energy: $\Delta \tilde{F}_{elec} = -\Delta U_{elec} = -\frac{\varepsilon-1}{8\pi}E^2 A_{cs}h$

**Step 3: Total Free Energy and Minimization**

The total change in the free energy of the system as a function of $h$ is the sum of the gravitational and electrical contributions: $\Delta \tilde{F}(h) = U_g + \Delta \tilde{F}_{elec} = \frac{1}{2}\rho g A_{cs}h^2 - \frac{\varepsilon-1}{8\pi}E^2 A_{cs}h$

To find the equilibrium height, we minimize this free energy by taking its derivative with respect to $h$ and setting it to zero: $\frac{d(\Delta \tilde{F})}{dh} = 0$ $\frac{d}{dh}\left(\frac{1}{2}\rho g A_{cs}h^2 - \frac{\varepsilon-1}{8\pi}E^2 A_{cs}h\right) = 0$
$\rho g A_{cs}h - \frac{\varepsilon-1}{8\pi}E^2 A_{cs} = 0$

**Step 4: Solve for h**

The cross-sectional area $A_{cs}$ is a common factor and cancels out. $\rho g h = \frac{(\varepsilon-1)E^2}{8\pi}$

Solving for the height $h$: $h = \frac{(\varepsilon-1)E^2}{8\pi\rho g}$

This final expression for $h$ uses only the symbols provided in the problem statement.

$$h = \frac{(\varepsilon - 1)E^2}{8\pi\rho g}$$

**SEED score:**
100

**DeepSeek-V3 Answer:**

To determine the height $h$ by which the liquid surface rises inside a vertical parallel-plate capacitor, we analyze the balance between the gravitational potential energy and the electrostatic energy.

**1. Gravitational Potential Energy:**

The work done against gravity to raise the liquid to height $h$ is given by the gravitational potential energy:

$$U_{\text{grav}} = \frac{1}{2}\rho g h^2$$

(The factor of $\frac{1}{2}$ arises because the center of mass of the raised liquid column is at height $\frac{h}{2}$.)

**2. Electrostatic Energy:**

When the liquid rises, it displaces air between the capacitor plates. The change in electrostatic energy due to the presence of the dielectric liquid is:

$$U_{\text{elec}} = -\frac{1}{2}\varepsilon_0(\varepsilon - 1)E^2 h$$

Here, $\varepsilon_0$ is the permittivity of free space, and $\varepsilon$ is the relative permittivity (dielectric constant) of the liquid. The negative sign indicates a reduction in energy when the dielectric liquid is introduced.

**3. Equilibrium Condition:** The system reaches equilibrium when the total free energy $\tilde{F} = U_{\text{grav}} + U_{\text{elec}}$ is minimized. Taking the derivative with respect to $h$ and setting it to zero:

$$\frac{d\tilde{F}}{dh} = \rho g h - \frac{1}{2}\varepsilon_0(\varepsilon - 1)E^2 = 0$$

Solving for $h$:

$$h = \frac{\varepsilon_0(\varepsilon - 1)E^2}{2\rho g}$$

Thus, the height by which the liquid surface rises is:

$$\boxed{h = \frac{\varepsilon_0(\varepsilon - 1)E^2}{2\rho g}}$$

**SEED score:**
36.5

**Comments:**
The model fundamentally misidentifies the energy involved. It calls a specific energy term "electrostatic energy," when that term actually represents the total energy change of the entire system, including the work done by the battery. This is a critical distinction in physics. After making that initial conceptual mistake, the model performs all the subsequent math—the calculus and algebra—perfectly. It followed the recipe's instructions (the math) correctly, but it used the wrong ingredient (the physics concept).

**Expert Comments:**
The model misunderstand the physical concept and the units required by the problem.

**Error Type:**
Concept and Model Misuse, Mathematical or Logical Errors

---

**DeepSeek-R1 Answer:**
To determine the height $h$ by which the liquid surface rises inside the vertical parallel-plate capacitor is determined by minimizing the free energy $\tilde{F}$ of the system, which is minimized when the potential between the plates is constant. The relevant symbols are provided: $h$, $\tilde{F}$, $\rho$, $g$, $\varepsilon$, $\pi$, and $E$.

The free energy $\tilde{F}$ includes the Gibbs free energy of the capacitor and the gravitational potential energy of the liquid. Under constant voltage, the Gibbs free energy for the capacitor is $-\frac{1}{2}CV^2$, where $C$ is the capacitance and $V$ is the voltage. The gravitational potential energy is $\frac{1}{2}\rho g w d h^2$, where $w$ is the width of the plates and $d$ is the plate separation. However, $w$ and $d$ cancel out in the minimization process.

The capacitance $C$ depends on the height $h$ of the liquid. The electric field $E$ is constant and uniform between the plates due to the constant voltage and the orientation of the dielectric interface. Specifically, $E = V/d$, and this field magnitude remains unchanged regardless of the liquid height because the voltage is constant.

In Gaussian units, the expression for $h$ is derived by minimizing $\tilde{F}$ with respect to $h$, leading to:

$$h = \frac{E^2(\varepsilon - 1)}{8\pi\rho g}$$

This result is consistent with the force balance approach, where the electrostatic force per unit area at the liquid-air interface balances the hydrostatic pressure due to the raised liquid column.

The symbols used are: $E$ (electric field magnitude in air when no liquid is raised), $\varepsilon$ (dielectric constant of the liquid, relative permittivity), $\rho$ (density of the liquid), $g$ (acceleration due to gravity), and $\pi$ (mathematical constant). The factor $8\pi$ arises from the use of Gaussian units in the formulation.

Thus, the height $h$ is given by the above expression.

$$\boxed{h = \frac{E^2(\varepsilon - 1)}{8\pi\rho g}}$$

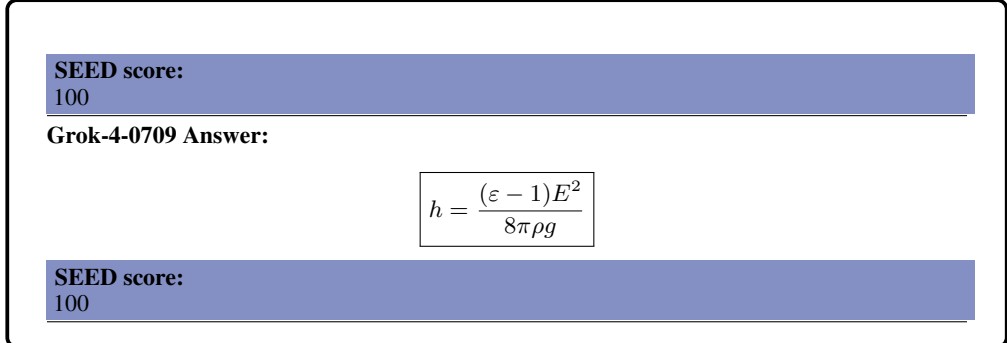

**SEED score:**
100

**Grok-4-0709 Answer:**

$$h = \frac{(\varepsilon - 1)E^2}{8\pi\rho g}$$

**SEED score:**
100

## F  USE OF LLMS

During the preparation of this work, the author(s) used LLMs to improve the language and readability. After using this tool/service, the author(s) reviewed and edited the content as needed and take(s) full responsibility for the content of the publication.

## G  FUTURE DIRECTION

The error categories we have defined in Sec. 4 and shown in Fig. 6 provide a clear guide for future research directions.

**Addressing "Concept and Model Misuse" through domain-specific fine-tuning.** Our finding that "Concept and Model Misuse" is the most frequent error type strongly motivates the use of our benchmark's source materials for domain-specific fine-tuning because this may largely come from insufficient understanding of background knowledge like physical concept and assumption. Creating a high-quality dataset from these graduate-level textbooks can impart the requisite foundational knowledge directly. Alternatively, these materials are perfectly suited for Retrieval-Augmented Generation (RAG), enabling the model to ground its reasoning in authoritative domain knowledge at inference time (Liu et al., 2024), thereby directly addressing the primary failure mode we observed.

**Mitigating "Mathematical or Logical Errors" with neuro-symbolic methods.** The high rate of these errorshighlights a well-documented limitation in the symbolic reasoning capabilities of current LLMs and this type of error may originate from the bottleneck of inference ability. This motivates using LLM $\rightarrow$ Symbolic approaches (Yang et al., 2025), like Program-Aided Language Models (PAL) (Gao et al., 2023) and Program of Thoughts (PoT) (Chen et al., 2022), where the LLM translates the problem into a formal language (like Python), and a deterministic symbolic engine handles the exact mathematical execution.

**Correcting "System Limitations" with instruction finetuning.** Failures in following output constraints can be directly addressed using our benchmark. The highly structured and consistent format of the ground-truth solutions in CMPhysBench makes it a perfect resource for instruction finetuning to better align models with specific task requirements (Wei et al., 2021; Wang et al., 2023).

Furthermore, unlike binary accuracy, SEED provides fine-grained, non-binary partial credit. This makes it an ideal dense reward signal for training paradigms like Reinforcement Learning with Verifiable Rewards (RLVR), allowing models to learn incrementally even from imperfect solutions (Gunjal et al., 2025) and transforming SEED from a static evaluation tool into a dynamic component for future model training.

## H  DETAILED INTERPRETATION OF SYMBOLS IN FIGURE 1

Given that Condensed Matter Physics (CMP) involves specialized terminologies that may lie beyond the general research scope, and acknowledging that the symbolic representations (such as the particle creation/annihilation operators and the specific variable shorthands used in the tree diagrams) in Figure 1 might be confusing to non-experts, we provide a comprehensive background reference here. The following table details the fundamental operators, physical parameters, and the specific shorthand

notations defined in the Anderson s-d exchange model problem. This supplement aims to bridge the gap for readers from different backgrounds and facilitate a clearer understanding of both the physics and the computational graph representation.

Table 6: Nomenclature and physical interpretations of notations used in Figure 1.

| Symbol | Physical Interpretation |
| --- | --- |
| **1. Fundamental Operators & Indices** | |
| $H$ | The **Hamiltonian** representing the total energy of the quantum many-body system. |
| $\sum_{k,\sigma}$ | Summation over all possible momenta ($k$) and spins ($\sigma$). |
| $\sigma/\bar{\sigma}$ | Electron spin index ($\uparrow$ or $\downarrow$). $\bar{\sigma}$ denotes the opposite spin of $\sigma$. |
| **2. Particle Operators (Second Quantization)** | |
| $C_{k\sigma}^{\dagger}/C_{k\sigma}$ | Creation/Annihilation operators for **conduction electrons** (the mobile electron sea). |
| $d_{\sigma}^{\dagger}/d_{\sigma}$ | Creation/Annihilation operators for the **impurity** electron (localized state). |
| $n_{d\sigma}$ | Number operator ($d_{\sigma}^{\dagger}d_{\sigma}$) counting the occupation of impurities. |
| **3. Energy & Interaction Parameters** | |
| $E_{k\sigma}/E_{d\sigma}$ | Energy levels for conduction electrons and impurity electrons, respectively. |
| $U$ | **Coulomb Repulsion**: The energy penalty for two electrons occupying the same impurity site. |
| $V_{kd}$ | **Hybridization**: The interaction strength allowing electrons to hop between the conduction band and the impurity. |
| $g_0/g_i$ | Landé g-factors (dimensionless magnetic moment) for electrons and impurities. |
| $h/\mu_B$ | External magnetic field and Bohr Magneton. |
| **4. Problem-Specific Shorthands (Variables in Tree Diagrams)** | |
| $\omega$ | The frequency (energy) variable in the complex plane, appearing as a leaf node in the expression tree. |
| $\langle\langle A\vert B\rangle\rangle_{\omega}$ | The Green's Function notation. It represents the correlation between state $B$ to state $A$. |
| $a_{k\sigma}$ | **Shorthand** for the mixed Green's function $\langle\langle C_{k\sigma}\vert d_{\sigma}^{\dagger}\rangle\rangle_{\omega}$. This variable appears explicitly in the final answer and the tree structure. |
| $b_{\sigma}$ | **Shorthand** for the impurity Green's function $\langle\langle d_{\sigma}\vert d_{\sigma}^{\dagger}\rangle\rangle_{\omega}$. Used to simplify the equation of motion. |

