# OpenReview forum: "CMPhysBench: A Benchmark for Evaluating Large Language Models in Condensed Matter Physics"
_ICLR.cc/2026/Conference — ICLR 2026 Poster_

### Official Review · Reviewer_TCqD · 2025-10-27

**Soundness:** 3
**Presentation:** 2
**Contribution:** 3
**Rating:** 6
**Confidence:** 2

**Summary:**

The paper introduces CMPhysBench, a benchmark designed to evaluate LLMs for Condensed Matter Physics (CMP) problems. The benchmark consists of 520 graduate-level calculation problems, spanning subfields such as magnetism, superconductivity, semiconductors, and strongly correlated systems. Compared to prior benchmarks, CMPhysBench emphasizes symbolic reasoning. A key contribution is the SEED metric, which extends existing expression-based similarity measures to support multiple answer types through syntax tree representations. The authors evaluate 18 proprietary and open-source LLMs, finding that even top models like Grok-4 achieve only 28% accuracy, suggesting a significant gap in domain-specific reasoning for CMP.

**Strengths:**

- The dataset is a good contribution, as it is manually curated by domain experts and spans diverse CMP subfields.

- SEED provides a novel measure of partial correctness, addressing a gap in symbolic assessment.

- The evaluation across 18 models is comprehensive, offering generalizability and comparative insight.

- Evaluation with human alignment validation and the error analysis with detailed categorization of reasoning failures helps to verify the findings.

**Weaknesses:**

- Although the error analysis is informative, the reasoning behind LLM failures remains speculative rather than causal.

- Maybe it's because CMP is a very specific area beyond my research scope. I find the dot examples in the figures very confusing. If a bit more background can be provided to explain the meaning of those symbols, it will help non-experts to better understand the paper.

**Questions:**

- Could the SEED metric be adapted to multimodal problems involving diagrams or figures in physics textbooks?

- Would you expect domain-specific fine-tuned LLMs (e.g., physics-trained models) to exhibit qualitatively different failure modes than general LLMs?

---

> ### Author Response · Authors · 2025-11-21
>
> We are very grateful for your positive and comprehensive review. Thank you for recognizing the value of our expert-curated dataset, the novelty of the SEED metric, our extensive model evaluation, and the rigor of our validation process. Your insightful questions and feedback are incredibly helpful for improving the paper's clarity and impact. We address your points below.
>
> 1. **On Error Analysis Being Descriptive Rather Than Causal (Weakness 1):**
>
>     We agree and have expanded our analysis with qualitative case studies to offer deeper insights. You have made an insightful point that our error analysis is primarily descriptive. Our main goal was to first **present a clear map of the types of errors** that are most prevalent. The eight error categories were summarized by domain experts through observation and reflect typical problems that also occur in human physics problem-solving, making them a robust framework for diagnosis.
>
>    While this map itself provides a crucial roadmap for future research, linking specific errors to targeted solutions as we detailed for Reviewer aQN2 (*e.g. Addressing "Concept and Model Misuse" through domain-specific fine-tuning*), we agree that a more causal analysis is a valuable direction. To offer deeper insights into *why* these failures occur, and in direct response to your feedback, we have expanded **Appendix E.3** to include a more detailed, qualitative analysis of several typical incorrect solutions. This provides readers with a concrete, case-study view of the models' flawed reasoning processes, directly complementing our quantitative results by moving closer to a causal understanding.
>
> 2. **On the Accessibility of "Dot Examples" for Non-Experts (Weakness 2):**
>
>    Thank you, we have added explanations in a new appendix to improve the paper's accessibility. This is invaluable feedback, and we apologize that the domain-specific examples in our figures were confusing. To make the paper more accessible to a broader audience, we have now added a dedicated section in the appendix **explaining the specialized physics notation and function used in our examples** (e.g., Figure 1). This provides the necessary context for non-expert readers to understand the examples without disrupting the main text.
>
> 3. **On Adapting the SEED Metric to Multimodal Problems (Question 1):**
>
>    **SEED is already applicable to most multimodal physics problems without modification.** This is an excellent and forward-looking question. **The key is that SEED's applicability is determined by the format of the final answer, not the problem statement.** Many physics problems that include diagrams (e.g., circuit diagrams, crystal lattice structures) still require a final answer that is symbolic or numerical. Since SEED is designed to robustly handle a wide range of structured answer types (Equation, Expression, Tuple, Numeric, etc.), it can be directly and effectively applied in these common multimodal scenarios. The more complex case where the model's required output is itself a diagram (e.g., "draw the Feynman diagram") would fall outside our current scope and require a different, vision-based evaluation paradigm.
>
> 4. **On the Expected Failure Modes of Domain-Specific LLMs (Question 2):**
>
>    This is a superb research question that gets to the heart of why benchmarks like ours are needed. We do expect domain-specific fine-tuned models to exhibit qualitatively different failure modes. Our hypothesis is: The frequency of **"Concept and Model Misuse"** errors would significantly decrease, as the model would have a much stronger grasp of core physical principles. However, errors related to fundamental reasoning, such as **"Mathematical or Logical Errors"** might persist, as they are likely more tied to the architectural limitations of current models in performing complex symbolic manipulation. A primary goal of CMPhysBench is to provide the platform to test exactly this kind of hypothesis, and we have added a brief note on this expected shift to our discussion section.
>
> Thank you once again for your thoughtful feedback, which will undoubtedly help us improve the quality and impact of our paper.

---

### Official Review · Reviewer_K4Jm · 2025-10-31

**Soundness:** 3
**Presentation:** 2
**Contribution:** 3
**Rating:** 6
**Confidence:** 4

**Summary:**

Graduate level condensed matter physics benchmark manually curated in high detail. The authors main contributions are (1) mix of LLM and expert question generation (2) presentation of a new metric SEED which helps to reward both partial correctness as well as (intended) to help with symbolic equivalence matching of answers, correlating well with human preference, (3) evaluation of several SoTA models on CMPhysBench showing that models currently struggle with condensed matter physics.

**Strengths:**

1, Scope and subject: Graduate level benchmark based on standard graduate textbooks, requiring complex step-by-step solutions across diverse answer types. Specifically, CMYPhysBench is a non MC/QA benchmark, so much more difficult.

2. Diversity and coverage: Clear balance between categories, and clear explanation and validation of the source material from which benchmark is derived from. The authors also perform strong analysis on failure modes, which are possibly actionable and of interest to members of the AI for science community trying to understand LM capabilities on scientific domains (and when to (or not) use models for specific scientific tasks).

3. SOTA LLMs moderately struggle on this benchmark making it of key interest to the community and interesting for marking LLM progress over time.

**Weaknesses:**

1. Relevance of SEED as an benchmark evaluation metric versus actual accuracy.

The goal seems to reward partial correctness (which is understandable from an RL or intermediate reward feedback perspective), however in practice: does SEED actually properly weight when LMs make minor incorrect reasoning steps (or does it only purely give partial credit when LMs fail to decode a final correct answer)? Some more discussion on this would be helpful.

Related, from model thinking trajectories, how well does SEED correlate with minor incorrect steps in reasoning: I’m worried that SEED partial correctness of answer may not correlate with minor reasoning incorrectness, which may defeat some of the benefit of this partial correctness.

Related, is it true that partial misses in edit distance should be only partially penalized just at an AST level?

2. Human preference seems to be binary (0 or 1): are there studies on how accurate human labelling is here? Were questions + ground truth answers verified by multiple raters? As such would the human grading here be close to an “accuracy” or a “preference” type of statistic?

**Questions:**

Figure 6 very difficult to read even while zoomed in (Model text and also the colors)

Figure 4: I’m unclear on how SEED is actually computed here, where does the 60 in Model Response 1 come from?

---

> ### Author Response · Authors · 2025-11-21
>
> We sincerely thank you for your detailed and constructive review. We are particularly grateful for your appreciation of the benchmark's graduate-level scope, its non-trivial open-ended format, the balanced coverage of topics, and the value of our failure analysis for the AI for science community. Your insightful questions are crucial and will help us significantly improve the clarity and impact of our work.
>
> You raised several important points regarding the SEED metric, our human evaluation process, and the presentation of figures. We address each in turn below.
>
> 1. **On SEED's Correlation with Reasoning Steps (Weakness 1):**
>
>    You raised a crucial point about whether SEED's partial credit for the final answer accurately reflects the correctness of the intermediate reasoning.
>
>    This is a central consideration in our work. We argue that in complex physics derivations, reasoning errors almost invariably propagate to the final symbolic expression. **A minor error often leads to a small structural change, while a major conceptual mistake results in a vastly different one.** This is precisely why, inspired by prior work like PHYBench, we use a metric that evaluates the final symbolic structure to award partial credit.
>
>    However, we agree this is not the full picture. That is why our methodology has a second, complementary component: **our detailed error analysis (Section 4.1) is designed specifically to evaluate the reasoning process itself.** By identifying failures like "Concept and Model Misuse," we directly diagnose flaws in the model's thinking trajectory. In the revised manuscript, we have clearified that SEED provides a scalable outcome-based score, while our error analysis offers a direct assessment of the reasoning steps.
>
> 2. **On the Details of Human Evaluation (Weakness 2):**
>
>    You asked for clarification on our human rating process and data verification.
>
>    - **Human Correlation was Binary:** To validate SEED's ability to function as a strict **accuracy** metric, we performed a direct comparison against binary expert judgments. For this specific analysis, human experts labeled answers as strictly correct (1) or incorrect (0). We then converted the SEED score into a corresponding binary value, where only a perfect score (SEED = 100) was considered correct (1). The high Spearman correlation ($\rho = 0.90$) we report is between these two binary sets, confirming that a perfect SEED score is a highly reliable indicator of what a human expert would deem a perfectly correct answer. And this clarification and settings have been added to the Section 4.3 of the revised manuscript.
>    - **Ground-Truth Verification:** Separately, to ensure the integrity of the benchmark itself, every question and its ground-truth solution underwent a rigorous vetting process. Each item was **independently reviewed by 2-3 domain experts**. For problems adapted from textbooks, our solutions were also cross-referenced with the original source materials to guarantee their correctness.
>
> 3. **On Figure Readability and SEED Calculation (Question 1 & 2):**
>
>    - **Figure 6 Readability:** We acknowledge that Figure 6 is difficult to read. We will redraw it with higher resolution, using a clearer color scheme and larger fonts to ensure all text is perfectly legible.
>
>    - **Figure 4 Readability:** Regarding the calculation in Figure 4, we regret not making its motivation explicit. The scoring function is a design choice adapted from the PHYBench [1] benchmark to award partial credit for answers with minor structural or coefficient errors. The score is calculated based on the relative edit distance $r$ between the ground-truth ($T_{gt}$) and the generated ($T_{gt}$) expression trees, as defined by the following function:
>      $$
>      r = \frac{\text{Distance}(T_{\text{gt}}, T_{\text{gen}})}{\text{Size}(T_{\text{gt}})}, \quad \text{score} = \begin{cases} 100, & \text{if } r = 0 \text{ (exact match)}, \\ 60 - 100r, & 0 < r < 0.6, \\ 0, & r > 0.6. \end{cases}
>      $$
>      This function assigns a full score of 100 for an exact match ($r=0$), linearly scales the score down from a baseline of **60** for partial correctness, and assigns a score of 0 for expressions that are significantly different $r> 0.6$ ). In the revised manuscript, we will add this formula and a clear explanation to the methodology section to fully justify our approach.
>
> We will explicitly detail these points in our revised methodology section to leave no ambiguity. We are confident these clarifications and revisions will fully address your concerns. Thank you once again for your valuable feedback.
>
> [1] Phybench: Holistic evaluation of physical perception and reasoning in large language models. In *NeurIPS*, 2025.

---

### Official Review · Reviewer_nemX · 2025-11-01

**Soundness:** 2
**Presentation:** 2
**Contribution:** 1
**Rating:** 4
**Confidence:** 3

**Summary:**

This paper presents CMPhysBench, a new benchmark to test LLMs on graduate-level Condensed Matter Physics. It's made of 520+ hard calculation problems from textbooks. They also created a new scoring metric called SEED that gives partial credit for complex math answers (like equations or tuples) using tree-based analysis. Their tests on 18 LLMs show that even the best models, like Grok 4, perform poorly (36 SEED score, 28.9% accuracy), showing a big gap in this specific domain.

**Strengths:**

The paper's main strength is tackling a new, hard domain: graduate-level condensed matter physics. Most benchmarks are easier, so this is a needed step up. The SEED metric is also a big plus; it's a smart way to give partial credit on complex math answers instead of just right/wrong. This metric seems useful for other science benchmarks too. The testing of 18 models is thorough, and the error analysis in Figure 6 gives a good breakdown of why models fail, with "Concept and Model Misuse" being the biggest problem.

**Weaknesses:**

The main weakness I see is in the error analysis. The authors used GPT-4o to categorize all the model mistakes. While this is fast, it's not clear how accurate GPT-4o is at this task. It would be better if they had human experts check a sample of these to confirm the error breakdown. Also, the SEED score focuses on the final boxed answer. The prompt asks for step-by-step solutions, but it's not clear if the steps themselves are evaluated. A model could get the right answer with the wrong steps.

**Questions:**

For the error analysis, how do you know GPT-4o's categorizations are correct? Did you have any humans double-check its work? Does your evaluation look at the reasoning steps, or just the final answer in the box? It seems possible for a model to get the right answer by luck or by making mistakes that cancel out.

---

> ### Author Response · Authors · 2025-11-21
>
> We thank you for your thorough review and insightful summary. We are grateful you recognized the strengths of our work, including the challenging new domain, the novelty and utility of the SEED metric, and the comprehensive nature of our model analysis.
>
> You raised two critical questions regarding our methodology, which we are happy to clarify: the reliability of GPT-4o for our error analysis, and our approach to evaluating reasoning steps versus the final answer.
>
> 1. **On the reliability of GPT-4o for error categorization:**
>
>    You raised a crucial question about the accuracy of using GPT-4o for error categorization and rightly suggested the need for human expert validation. We appreciate this scrutiny and would like to clarify the rigorous process we undertook, which we regret not detailing sufficiently in our initial submission.
>
>    - Our decision to explore an automated analysis was **inspired by recent successful efforts** in LLM evaluation, such as xVerify [1] and MathVerse [2], which have demonstrated the potential of using powerful models (like GPT-4o) for fine-grained assessment. To adapt this approach reliably to our highly specialized domain, two co-authors with expertise in condensed matter physics first **summarized common error types** and then manually **annotated** 300 diverse question-response pairs.
>
>    - **Our automated method achieved a 98% agreement rate with human expert consensus.** We iteratively refined our GPT-4o prompt, finding that a Chain-of-Thought (CoT) approach was key to high accuracy. The final validation against our 300-sample expert set gave us strong confidence that our method is a **valid and scalable proxy for expert evaluation**, as detailed in the revised `Section 4.1`.
>
> 2. **On evaluating reasoning steps versus the final answer:**
>
>    You correctly pointed out that the SEED score evaluates the final boxed answer and wondered if we assess the intermediate reasoning steps, raising the valid concern that a model could get the right answer for the wrong reasons. **Our evaluation framework is a two-part system that assesses both the final answer's correctness and the reasoning process.** These two components are designed to be complementary.
>
>    - **The SEED score provides a objective measure of the final answer.** At the graduate physics level, the chance of arriving at a precisely correct complex expression through flawed reasoning is **exceedingly low**. Therefore, a correct final answer is a very strong indicator of a correct reasoning process, and SEED’s partial credit often correlates with the severity of the reasoning error.
>    - **Our error analysis focuses specifically on diagnosing flaws in the reasoning steps.** The error categories we present (e.g., "Concept and Model Misuse" or "Mathematical Errors") can **only be identified by manually scrutinizing the entire step-by-step solution**. This analysis is our dedicated evaluation of the model's reasoning process, explaining *how* a model failed, not just that it failed.
>
>    In summary, we use a dual approach: the SEED score provides a robust, quantitative, and scalable measure of performance, while our human-validated error analysis provides a deep, qualitative diagnosis of the reasoning failures. We will clarify this dual-evaluation strategy in our methodology section to make the distinct and complementary roles of these two components much clearer.
>
> We are confident that these clarifications and additions will fully address your concerns and better reflect the rigor of our evaluation methodology. Thank you once again for your feedback.
>
> [1] xverify: Efficient answer verifier for reasoning model evaluations. In *arXiv*, 2025.
>
> [2] Mathverse: Does your multi-modal llm truly see the diagrams in visual math problems? In *ECCV*, 2024.

---

### Official Review · Reviewer_aQN2 · 2025-11-01

**Soundness:** 3
**Presentation:** 3
**Contribution:** 2
**Rating:** 6
**Confidence:** 2

**Summary:**

This manuscript introduces CMPhysBench, a novel benchmark that successfully establishes a rigorous standard for evaluating LLMs on advanced scientific reasoning tasks in physics. The adoption of a fine-grained evaluation protocol is a major contribution.

**Strengths:**

1. High Problem Difficulty

Focuses exclusively on graduate-level material, comprising more than 520 meticulously curated questions that require LLMs to generate complete, step-by-step solutions for complex calculation problems. This moves beyond the limitations of high school or undergraduate benchmarks, demanding advanced mathematical rigor and conceptual understanding.

2. Expert-Aligned Metric

The proposed Scalable Expression Edit Distance (SEED) score provides highly accurate, fine-grained, non-binary partial credit for mathematical responses. SEED exhibits the highest correlation (ρ = 0.90) with human expert ratings, demonstrating superior alignment in evaluating complex symbolic reasoning compared to prior metrics.

3. Extensive Model Analysis

The paper conducts a comprehensive empirical study evaluating 18 proprietary and open-source LLMs. This extensive analysis identifies a significant capability gap, with the best models achieving only a 36 average SEED score and 28% accuracy, providing quantitative illumination of specific failure modes across the LLM ecosystem.

**Weaknesses:**

1. It would be better if the authors could discuss how the issues of LLM in this domain identified in the analysis could be mitigated in the future research. Currently, the analyses only show LLM can make multiple types of error and it is still unclear how to improve LLM to avoid such errors. Proposing potential solutions for the identified errors could further improve the contribution of the paper.

**Questions:**

1. please add additional analyses or discussions towards future research directions or the contribution of this paper towards the future improvements of LLM in this domain.

---

> ### Author Response · Authors · 2025-11-21
>
> We thank the reviewer for their positive feedback and for highlighting the opportunity to elaborate on future research directions. We have incorporated this discussion into *Appendix* of the revised manuscript and summarize the key points below:
>
> 1. **Our detailed error analysis provides a direct roadmap for targeted model improvements.**
>    - **Addressing "Concept and Model Misuse" through domain-specific fine-tuning.** Our finding that **"Concept and Model Misuse"** is the most frequent error type strongly motivates the use of our benchmark's source materials for **domain-specific fine-tuning** because this may largely come from insufficient understanding of background knowledge like physical concept and assumption. Creating a high-quality dataset from these graduate-level textbooks can impart the requisite foundational knowledge directly. Alternatively, these materials are perfectly suited for **Retrieval-Augmented Generation (RAG)**, enabling the model to ground its reasoning in authoritative domain knowledge at inference time [1], thereby directly addressing the primary failure mode we observed.
>    - **Mitigating "Mathematical or Logical Errors" with neuro-symbolic methods.** The high rate of these errorshighlights a well-documented limitation in the symbolic reasoning capabilities of current LLMs and this type of error may originate from the bottleneck of inference ability. This motivates using **LLM → Symbolic approaches** [2], like Program-Aided Language Models (PAL) [3] and Program of Thoughts (PoT) [4], where the LLM translates the problem into a formal language (like Python), and a deterministic symbolic engine handles the exact mathematical execution.
>    - **Correcting "System Limitations" with instruction finetuning.**  Failures in following output constraints can be directly addressed using our benchmark. The highly structured and consistent format of the ground-truth solutions in CMPhysBench makes it a perfect resource for **instruction finetuning** to better align models with specific task requirements [5,6].
>
> 2. **The proposed SEED metric serves as a powerful signal for advanced model training.**
>
>    Unlike binary accuracy, SEED provides fine-grained, non-binary partial credit. This makes it an ideal **dense reward signal** for training paradigms like **Reinforcement Learning with Verifiable Rewards (RLVR)**, allowing models to learn incrementally even from imperfect solutions [7] and transforming SEED from a static evaluation tool into a dynamic component for future model training.
>
> By framing our findings in this constructive manner, we now better articulate how our benchmark serves not only to diagnose current limitations but also to actively guide the development of more capable and reliable scientific AI. Thank you once again for your insightful feedback, which has helped us strengthen the contribution of our paper.
>
> [1] How much can rag help the reasoning of llm. In arXiv, 2024.
>
> [2] Neuro-symbolic artificial intelligence: towards improving the reasoning abilities of large language models. In *IJCAI*, 2025.
>
> [3] Pal: Program-aided language models. In *ICML*, 2023.
>
> [4] Program of thoughts prompting: Disentangling computation from reasoning for numerical reasoning tasks. In *TMLR*, 2023.
>
> [5] Finetuned language models are zero-shot learners. In *arXiv*, 2021.
>
> [6] Self-instruct: Aligning language models with self-generated instructions. In *ACL*, 2023.
>
> [7] Rubrics as rewards: Reinforcement learning beyond verifiable domains. In *arXiv*, 2025.

---

### Author Response · Authors · 2025-11-26
**Waiting for Further Discussion**

Dear Reviewer,

Thank you again for your detailed and thoughtful review. We have provided a detailed response and revised the manuscript to address your comments, which has helped us significantly improve the paper. We welcome any further feedback you might be willing to share. Your insights are invaluable to us, and we're eager to address any remaining issues.

Thank you again for your support in improving our work.

Best regards,

All authors of CMPhysBench

---

### Meta-Review · Area_Chair_QmnZ · 2026-01-13

**Summary:**

This paper introduces CMPhysBench, a benchmark of 520+ graduate-level condensed matter physics calculation problems, requiring models to produce full solutions, and proposes SEED, a tree-based expression edit distance metric that gives fine-grained partial credit for symbolic answers.

Reviewers broadly agree the benchmark fills a real gap (graduate/research-level CMP), the scale and coverage are strong, and SEED is a meaningful contribution with good reported alignment to expert judgments. Main concerns were about (i) how SEED relates to reasoning quality vs just final answers, (ii) the reliability of automated error analysis (GPT-4o labeling), and (iii) readability/accessibility for non-experts. The rebuttal adds a human validation study for the automated analysis, clarifies a dual-evaluation setup (SEED for outcome + taxonomy for reasoning), improves figures/notation explanations, and expands discussion on mitigation directions.

**Reviewer Concerns:**

Addressed by rebuttal:
- Automated error analysis reliability: Authors report a 300-sample expert-labeled validation with very high agreement for GPT-4o-based categorization.
- Outcome vs reasoning evaluation: Clarified that SEED scores final symbolic correctness while the error taxonomy diagnoses reasoning failures; added the SEED formulation details.
- Actionability / future directions: Added mitigation discussion mapping error types to potential approaches (domain finetuning/RAG, neuro-symbolic methods, instruction tuning, RL with dense rewards).
- Readability: Improved figure legibility and added a notation primer / explanations for domain-specific symbols.

Still outstanding / limitations:
- SEED as “partial credit” vs scientific correctness: Some reviewers remain uneasy about whether AST-level partial credit always matches meaningful progress in reasoning; the rebuttal helps but the concern is inherent.
- Causality of failure analysis: Even with case studies, the error analysis is still mostly diagnostic rather than truly causal.
- Human alignment details: The “binary expert judgment” alignment is clarified, but it doesn’t fully settle questions about how partial-credit SEED corresponds to nuanced human grading.

**Reviewer Scores:**

Likely post-discussion stances:
1. aQN2: 6 → 6 (request for mitigation directions addressed).
2. nemX: 4 → 5–6 (key worry about GPT-4o labeling addressed by human validation + clearer dual-eval framing).
3. K4Jm: 6 → 6 (SEED and figure clarity concerns addressed; likely remains a solid accept).
4. TCqD: 6 → 6 (accessibility and “why failures” concerns partially addressed).

---

### Decision · Program_Chairs · 2026-01-26

Accept (Poster)